

# Glacier change and glacial lake outburst flood risk in the Bolivian Andes

Simon J. Cook[1,2,*], Ioannis Kougkoulos[1,2], Laura A. Edwards[2,3], Jason Dortch[2,3], Dirk Hoffmann[4]

[1]School of Science and the Environment, Manchester Metropolitan University, Chester Street, Manchester, M1 5GD, UK.

[2]Cryosphere Research at Manchester (CRAM), Oxford Road, Manchester, UK.

[3]Department of Geography, University of Manchester, Oxford Road, Manchester, M13 9PL, UK.

[4]Bolivian Mountain Institute, Casilla 3-12417, La Paz, Bolivia.

*Correspondence to*: Simon J. Cook (S.J.Cook@mmu.ac.uk)

**Abstract.** Glaciers of the Bolivian Andes represent a vital water resource for Andean cities and mountain communities, yet relatively little work has assessed changes in their extent over recent decades. In many mountain regions, glacier recession has been accompanied by the development of proglacial lakes, which can pose a glacial lake outburst flood (GLOF) hazard. However, no studies have assessed the development of such lakes in Bolivia despite recent GLOF incidents here. Our mapping from satellite imagery reveals an overall areal shrinkage of $228.1 \pm 22.8$ km$^2$ (43.1%) across the Bolivian Cordillera Oriental between 1986 and 2014. Shrinkage was greatest in the Tres Cruces region (47.3%), followed by the Cordillera Apolobamba (43.1%) and Cordillera Real (41.9%). A growing number of proglacial lakes have developed as glaciers have receded, in accordance with trends in most other deglaciating mountain ranges, although the number of ice-contact lakes has decreased. The reasons for this are unclear, but the pattern of lake change has varied significantly throughout the study period, suggesting that monitoring of future lake development is required as ice continues to recede. Ultimately, we use our 2014 database of proglacial lakes to assess GLOF risk across the Bolivian Andes. We identify 25 lakes that pose a potential GLOF threat to downstream communities and infrastructure. We suggest that further studies of potential GLOF impacts are urgently required.

## 1 Introduction

Tropical glaciers are sensitive indicators of climate change and provide important information on climatic trends in locations where meteorological observations are sparse (Kaser, 1999; Vuille et al., 2008; Soruco et al., 2009). In this study, we focus on glacier change in the Bolivian Andes, which contains ~20 % of the world's tropical glaciers (Jordan, 1991, 1998; Kaser, 1999). Here, glacial meltwater is a vital water resource for major cities, such as La Paz and El Alto, as well as for mountain communities across the region (Vergara et al., 2007; Vuille et al., 2008; Oxfam, 2009; Rabatel et al., 2013; Rangecroft et al., 2013). La Paz and its neighbouring city of El Alto are home to ~2.3 million people, and it is estimated that glaciers supply ~15 % of the potable water supply to these areas. La Paz, like many Andean cities, derives some of its electricity from hydropower generation, which depends to some extent on glacial meltwater generation. Some researchers have expressed concern that




power generation during the dry season will become unreliable due to low water flows (Painter, 2007; Chevallier et al., 2011; Kaenzig, 2015), although this requires quantitative study.

There is also some evidence that more isolated mountain communities in Bolivia are suffering increasingly from the adverse effects of glacier recession and changing meltwater supply in response to climatic warming (Oxfam, 2009), although it does

not yet appear to be a direct driver of rural-to-urban migration (Kaenzig, 2015). Bolivia is the poorest country in South America and hence is very vulnerable to the impacts of climate change (Andersen and Verner, 2009; Winters, 2012). Indeed, it is estimated that only 56 % of Bolivia's rural population have access to safe water (Jeschke et al., 2012; Rangecroft et al., 2013), meaning that the sustainability of glaciers is a significant concern in the broader context of poverty and vulnerability to climate change.

Despite the regional importance of Bolivian glaciers, research to monitor their extent and response to climate change has been rather limited. Detailed mass balance and modelling studies have been performed for a few glaciers, such as the Zongo glacier (e.g. Sicart et al., 2011; Reveillet et al., 2015; Soruco et al., 2015). Other studies have documented the demise of Chacaltaya glacier, which disappeared in 2009 (Ramirez et al., 2001; Soruco et al., 2015). At a broader scale, Jordan et al. (1980) and Jordan (1991, 1998) developed the first inventory of glaciers in Bolivia. Soruco et al. (2009) calculated a volumetric reduction

of 43 % for 21 glaciers of the Cordillera Real over the period 1963 to 2006 (location shown in Figure 1b), whilst Albert et al. (2014) demonstrated that glaciers of the small range of Tres Cruces had lost approximately half of their surface area between 1975 and 2009 (locations shown in Figure 1b). However, a comprehensive quantification of glacier change across the Bolivian Andes has hitherto not been undertaken, and little is known about broad-scale glacier change in the last decade since the 1963-2006 study period of Soruco et al. (2009).

A crucial further issue for consideration is the development of potentially dangerous glacial lakes. Glaciers tend to erode subglacial basins and deposit eroded materials around their margins as lateral-frontal terminal moraines (Cook and Swift, 2012). Recession into these basins and behind impounding moraines causes meltwater to pond as proglacial and supraglacial lakes (Carrivick and Tweed, 2013; Cook and Quincey, 2015). In many mountain ranges, these lakes represent a glacial lake outburst flood (GLOF) hazard, as moraine dam integrity reduces over time, leading to dam failure, and as mass movements of

ice, snow and rock from surrounding valley slopes impact lakes, leading to wave-overtopping of moraine and bedrock dams (Clague and Evans, 2000; Richardson and Reynolds, 2000; Hubbard et al., 2005; Westoby et al., 2014). GLOF hazards have received almost no research attention in the Bolivian Andes, yet Hoffmann and Wegenmann (2013) documented a recent GLOF at Keara in the Apolobamba region in 2009, indicating that the potential impacts of such hazards have been overlooked. In this case, an ice-dammed lake burst, killing several farm animals, destroying cultivated fields, and washing away a road that

left the village population cut off for several months. Fortunately, no human casualties occurred. In other locations, several studies have shown that glacier recession has been accompanied by an increase in the number and size of proglacial lakes (e.g. Carrivick and Tweed, 2013; Carrivick and Quincey, 2014; Komori, 2008; Loriaux and Casassa, 2013; Hanshaw and Bookhagen, 2014; López-Moreno et al., 2014; Wang et al., 2014), raising concerns that glaciated mountain regions are becoming more hazardous with respect to GLOFs. As yet, however, no study has quantified proglacial lake development in



the Bolivian Andes to assess whether such a trend is prevalent here. Furthermore, no studies have assessed the extent to which these lakes represent a GLOF threat to downstream communities and infrastructure within Bolivia.

This study has three primary objectives to address these shortcomings: (1) to quantify glacier change in recent decades (from 1986 to 2014) across the Bolivian Andes; (2) to evaluate the development of proglacial lakes through this period; and (3) to

provide a first-pass assessment of whether any existing proglacial lakes represent a GLOF hazard that may require further monitoring. The time period is chosen based on the availability of satellite imagery, plus a desire to extend the period of observations from previous work on glacier change in the region – the observations of Soruco et al. (2009) extended to 2006 for the Cordillera Real, and those of Albert et al. (2014) extended to 2009 for the Tres Cruces region. The spatial extent of our study is chosen in order to provide a first integrated assessment of glacier change across the whole of the Bolivian Cordillera

Oriental.

## 2 Study region and methodology

### 2.1 Study region

Figure 1 shows a map of Bolivia with the footprint of the satellite imagery used to map glacier change from 1986 to 2014. This study focuses on the glaciated areas of the Cordillera Oriental of Bolivia, which itself can be divided into a series of

ranges. The most significant ice cover is in the Cordillera Real, in the middle part of our study region, and the Cordillera Apolobamba in the north, which straddles the Bolivia-Peru border. There are a series of smaller glaciated ranges further to the south of the main glaciated area of the Cordillera Real (including Huayna Potosí, Mururata, Illimani), and we consider all of these together as the Cordillera Real (after Jordan, 1991, 1998). The southernmost Tres Cruces region holds another small, glaciated range (Figure 1). We follow previous studies (Jordan, 1991, 1998; Soruco et al., 2009; Albert et al., 2014) in mapping

across the three regions covered by the three satellite footprints: Cordillera Apolobamba, Cordillera Real, Tres Cruces. Andean glaciers have experienced recent recession in response to increasing temperatures (e.g. Kaser, 1999; Vergara, 2007; Vuille et al., 2008; Rangecroft et al., 2013). Bolivian glaciers, unlike glaciers outside of the Tropics, accumulate mass during the summer wet season (November to April), and experience enhanced ablation during the winter dry season (May to October) and inter-seasonal periods when solar radiation is more intense (Jordan, 1998). Andean glaciers are particularly sensitive to

the El Niño Southern Oscillation (ENSO) (e.g. Francou et al., 1995, 2000; Wagnon et al., 2001). El Niño years lead to enhanced mass loss, whereas La Niña years tend to lead to reduced mass loss or even mass gains (Vuille et al., 2008).

### 2.2 Mapping glacier and lake change

Landsat satellite imagery, with a spatial resolution of 30 m, was obtained from the United States Geological Survey (USGS) using the Earth Explorer interface (http://earthexplorer.usgs.gov/). Data were obtained for the three footprints outlined in

Figure 1 for the years 1986, 1992, 1999, 2010 and 2014 (Table 1). Wherever possible, images were selected with minimal cloud cover, and all images were selected from the dry season when snow cover would have been at a minimum (which could



be confused for glacier ice). Ice cover was identified automatically in the first instance by using the TM3/TM5 band ratio (i.e. the ratio of red to short wave infrared) (e.g. Bolch et al., 2010) on atmospherically corrected imagery (Burns and Nolin, 2014). Some manual editing of the TM3/TM5 output was also required. Firstly, lakes that had been misidentified as ice cover were removed. Secondly, all glacier polygons smaller than 0.05 km$^2$ were removed as they probably represented snow patches rather
than glacier ice (Bolch et al., 2010). Thirdly, the areas identified as ice cover were checked against imagery in Google Earth, which has historic imagery stretching back to 2002 for much of the area considered in this study. Any other misidentified features (e.g. snow patches, areas of moraine or debris-covered ice, etc.) were then edited manually.

We estimated uncertainty in our mapping following the method outlined by Hanshaw and Bookhagen (2014). They assume that mapping errors are Gaussian, i.e. that ~68 % (1σ) of pixels will be subject to error. Uncertainty is calculated as:

$$Error\ (1\sigma) = \ (P/G) \ \cdot \ 0.6872 \ \cdot \ G^2/2 \tag{1}$$

Where $P$ is the measured glacier perimeter, and $G$ is the grid cell size. Uncertainties calculated using Equation 1 are ~10 %. Paul et al. (2013) found that error for measuring clean-ice glaciers, such as in this study, are on the order of ~5 %.

Lake extents were digitised manually with reference to both Landsat false colour composites, and NDWI (Normalised Difference Water Index) rasters (Huggel et al., 2002; Bolch et al., 2010). All visible lakes were mapped where they occurred
within approximately 2 km of the glacier margins. Ultimately, however, we focussed on those lakes located within 500 m of the glacier margins because these lakes are likely to be of most relevance in terms of GLOF risk (see below). As with glacier mapping, we estimate that our lake mapping uncertainty is ~10%.

## 2.3 Assessing glacial lake outburst flood risk

Several studies have proposed schemes or criteria by which to identify potentially dangerous glacial lakes, or to assess the
consequences of GLOF events (e.g. Allen et al., 2009; Huggel et al., 2004; McKillop and Clague, 2007; Bolch et al., 2008). In this study, we first identified lakes from the 2014 imagery that have the potential to burst by using some of the criteria outlined by Bolch et al. (2008); we then undertook a first pass assessment of the potential severity of the resultant flood event (i.e. by estimating flood peak discharge) following procedures outlined by Huggel et al. (2004) and McKillop and Clague (2007) where possible. Some data identified in these studies (such as moraine height, glacier velocity, etc.) were not available
to us and so could not be integrated into the risk assessment at this stage.

We considered potential GLOF sources to be (1) lakes in direct contact with glaciers; and (2) lakes within 500 m of the glacier margin that were also within 500 m of a steep (45° or steeper) slope. Ice-contact lakes could be directly affected by glacier calving events that generate waves with the potential to overtop impounding moraine and rock basin slopes. Previous studies have considered lakes within 500 m of a glacier to be potential GLOF sources (e.g. Wang et al., 2011; Wang et al., 2015).
Both ice-contact lakes and lakes within 500 m of a glacier could be impacted by ice and snow avalanches, which could also generate overtopping waves. These immediate proglacial areas are also affected by slope debuttressing associated with recent deglaciation, and hence are more likely to experience paraglacial slope failures that could impact the lakes (e.g. Hubbard et



al., 2005; Cook et al., 2013; Haeberli et al., 2016). Bolch et al. (2008) considered slopes steeper than 45° to be particularly hazardous in this regard. Hence, we generated a slope map from a 30 m resolution ASTER GDEM2 (downloaded from http://reverb.echo.nasa.gov/reverb/) to identify slopes of 45° or steeper that could shed material into proglacial lakes. Runout distances of ice and snow avalanches, rockfalls, rock avalanches, and debris flows vary widely, but appear to cluster on the

order of $10^2$ to $10^3$ m (Alean, 1985; Rickenmann, 1999, 2005; Copons et al., 2009). In the absence of detailed modelling of mass movement runout distances, we considered any proglacial lake within 500 m of a 45° or steeper slope to be potentially dangerous.

Next, we removed lakes that were unlikely to yield large flood events if they were to burst. This assessment was based principally on lake area, assuming that the smaller the area, the less significant the potential flood volume. Unfortunately, there

are no size criteria to determine the smallest lake size to include in any inventory of GLOF risk. We know that the damaging flood at Keara (Hoffmann and Wegenmann, 2013) resulted from the almost complete drainage of an ice-dammed lake that had a surface area of ~34,000 $m^2$. Hence, we set a lower lake size threshold of 30,000 $m^2$ to capture lakes of similar size to the one at Keara. This should be sufficient to capture all potentially dangerous lakes because all of the mapped lakes are either moraine-dammed, or sit within glacially overdeepened rock basins, and hence are less likely to drain completely as was the case for the

ice-dammed lake at Keara.

Having identified all of the lakes with a potential outburst risk, we then identified all such lakes within that population that posed a risk of damage to human interests (e.g. homes, roads or infrastructure, cultivated fields, etc.). For the most part, this was achieved using the GIS database of roads and settlements available freely from GeoBolivia (http://geo.gob.bo/). We also cross-referenced this with a visual assessment of human features in Google Earth and Bing Maps, plus our own observations

of a few of the sites. As a first assessment, we judged lakes to be dangerous where they had a direct hydrological connection to downstream infrastructure and communities (e.g. where a road or village was in the direct path of a GLOF event), and in general, we searched for such infrastructure within ~20 km downstream of the proglacial lakes (the complete drainage of the ice-dammed lake at Keara affected the channel for ~10 km downstream). Mapping of hydrological connectivity was achieved using the hydrological (i.e. flow routing) tools within ArcGIS 10.2.2, with the ASTER GDEM2 as input data.

We used Google Earth imagery to assess whether the lakes identified from previous steps were moraine- or rock-dammed. Moraine-dammed lakes are considered more dangerous because an initial trigger event, such as an avalanche-induced wave, could lead to breach incision in the moraine, and hence enhanced drainage of the lake (e.g. Westoby et al., 2015). Lakes sat within rock basins are less likely to experience breach incision. However, catastrophic drainage of lakes with bedrock dams have been noted in other tectonically active orogens (e.g. Dortch et al., 2011). Thus, both moraine- and rock-dammed lakes

represent potential sources of GLOFs.

To assess the severity of GLOF events from our inventory of potentially dangerous lakes, we first estimated lake volumes based on a measurement of their surface area. Cook and Quincey (2015) reviewed the empirical approaches that have been adopted in previous studies to model lake volume. Perhaps the most popular formula, which is based on a combination of data from ice-dammed, moraine-dammed, and thermokarst lakes, has been that of Huggel et al. (2002):



$$V = 0.104 \, A^{1.42} \tag{2}$$

Where V is lake volume in m³, and A is lake surface area in m². Cook and Quincey (2015) noted that the relationship of Huggel et al. (2002) performed well in estimating lake volumes in most cases, but that there were some situations where lakes could be especially shallow or deep, giving unusually small or large volumes for a given area. Hence, they advocated that geomorphological context be considered when deciding on which empirical approach to adopt. All of the lakes identified in our inventory are either moraine-dammed, or sit within rock basins. The relationship of Huggel et al. (2002) was shown by Cook and Quincey (2015) to perform well for estimating the volume of moraine-dammed lakes because the data used to generate Equation 1 were derived largely from moraine-dammed lakes. For those cases, we adopt Equation 1, but also use the larger dataset from Cook and Quincey (2015) to derive an empirical relationship specific to moraine-dammed lakes of a similar area range to those found in this study. This takes the form:

$$V = 0.097A^{1.4375} \tag{3}$$

We are not aware of any empirical formula for volume estimation where lakes are situated within rock basins. In the absence of any such formula, we use Equations 1 and 2 to provide a first order estimation of their volumes.

Lake volume can be used to estimate peak discharge ($Q_{max}$). Huggel et al. (2002) collated several empirical models for estimating GLOF peak discharge from lake volume, but ultimately adapted the relationship of Haeberli (1983) to give:

$$Q_{max} = \frac{2V}{t} \tag{4}$$

Where t, time, is equal to 1000 seconds. We used equation 3 to estimate peak discharge for the lakes identified as being potentially dangerous.

## 3 Results

### 3.1 Glacier change 1986-2014

Our results reveal that total glacier areal cover across the Bolivian Cordillera Oriental in 1986 was 529.3 ± 52.9 km², and that by 2014 this area had reduced to 301.2 ± 30.1 km² (Figure 2a). This represents a total areal reduction of 43.1 % over the 28-year study period. If the Peruvian Cordillera Apolobamba glaciers are included in the dataset, then the total glacier cover in 1986 was 626.5 ± 62.7 km² and in 2014 it was 351.7 ± 35.2 km², representing a 43.9 % reduction. Figure 2a illustrates the reduction in overall glacier cover across this period. Rates of ice loss appear to vary across the study period, with an initially rapid shrinkage between 1986 and 1992 (14.5 km² a⁻¹), relatively modest losses between 1992 and 1999 (5.1 km² a⁻¹), strong ice shrinkage between 1999 and 2010 (8.1 km² a⁻¹), and modest losses between 2010 and 2014 (4.0 km² a⁻¹) (except for the Tres Cruces region).

For consistency with earlier studies, we present results in Figure 2 of glacier areal change for separate glaciated mountain ranges, and Figures 3 to 5 illustrate glacier change as a series of maps for each region. All mountain ranges show decreases in



overall glacier area across the study period with a total loss of 43.1 % glacier cover in the Bolivian Cordillera Apolobamba ($172.3 \pm 17.2$ km$^2$ to $96.0 \pm 9.6$ km$^2$), 41.9% across the Cordillera Real ($315.2 \pm 31.5$ km$^2$ to $183.1 \pm 18.3$ km$^2$), and 47.3 % in the Tres Cruces region ($41.8 \pm 4.2$ km$^2$ to $22.0 \pm 2.2$ km$^2$). If Peruvian Cordillera Apolobamba glaciers are included, then the total loss for the Cordillera Apolobamba from 1986 to 2014 is 45.6 % ($269.5 \pm 27.0$ km$^2$ to $146.3 \pm 14.6$ km$^2$).

## 3.2 Proglacial lake development 1986-2014

Figure 6 illustrates how the number and areal cover of ice-contact and proglacial lakes has developed between 1986 and 2014 across the three regions of the Bolivian Andes. Data are presented for ice-contact lakes (Figure 6a and b), and for lakes within 500m of the 1986 ice margin (Figure 6c and d), which illustrates the cumulative change in proglacial lake number and area as the ice receded to its 2014 position. Figure 7 shows the total change in lake number and area across the Bolivian Andes.

Fig 6a indicates that that there is no clear pattern in the number of ice-contact lakes that developed from 1986 to 2014, although there has been an overall decline in the number of ice-contact lakes for all three regions. The Tres Cruces region shows the clearest trend of decline, although there were very few ice-contact lakes here throughout the study period (4 in 1986, and 0 in 2014). For the Cordillera Apolobamba, the number of lakes increased from 4 to 10 lakes between 1986 and 1999, before declining to 1 lake in 2014. The Cordillera Real region experienced the reverse situation to the Cordillera Apolobamba between 1986 and 2010, with a decline from 15 to 8 ice-contact lakes between 1986 to 1999, an increase to 12 lakes in 2010, before falling to 6 lakes in 2014. Figure 7a illustrates the overall decline in the number of ice-contact lakes across all regions, from 23 in 1986 to 7 in 2014.

Figure 6b shows the change in area of ice-contact lakes for all three regions. The trend in ice-contact lake area for the Cordillera Apolobamba and Tres Cruces region follows the trend in the number of ice-contact lakes shown in Figure 6a. The Cordillera Real experienced an overall 22 % increase in ice-contact lake area from $0.9 \pm 0.09$ km$^2$ in 1986 to $1.1 \pm 0.11$ km$^2$ in 2014, although peaked at $1.2 \pm 0.12$ km$^2$ in 2010. This increase in lake area, even though lake number has fallen across the same period, is driven by the growth of a few large ice-contact lakes (e.g. Laguna Glaciar at the northern tip of the Cordillera Real, and the large lake, Laguna Arkhata, beneath the summit of Mururata – Figure 4). Figure 7b shows the overall trend in ice-contact lake area across the study period, indicating a very slight (10 %) overall increase from $1.0 \pm 0.1$ km$^2$ in 1986 to $1.1 \pm 0.11$ km$^2$ in 2014 (represented mostly by lakes in the Cordillera Real), with a peak at $1.6 \pm 0.16$ km$^2$ in 2010. The number of ice-contact lakes stayed relatively stable from 1992 to 2010 (Figure 7a), yet the total area covered by these lakes increased (Figure 7b). This is explained by growth of ice-contact lakes until 2010, followed by detachment, leading to an overall decrease in both lake number and area by 2014.

Figure 6c shows that the number of proglacial lakes has increased from 1986 to 2014, for all regions, as the ice has drawn back from its 1986 position. These lakes fall within 500 m of the 1986 margin, and hence represent the cumulative total of lakes for the three regions as glaciers have receded. The greatest number of lakes exist in the Cordillera Real, which saw a 47 % increase from 92 lakes in 1986 to 135 lakes in 2014. The Tres Cruces region saw a 67 % increase from 24 lakes in 1986 to 40 lakes in 2014. The Cordillera Apolobamba has seen an overall increase of 72 % increase from 29 to 50 lakes across the study period,



although there was a peak in 2010 at 53 lakes. Figure 7a reveals that there has been an overall total increase of 55 % in proglacial lakes from 145 to 225 lakes.

Figure 6d shows that the area covered by lakes within 500 m of the 1986 ice margin has increased, and broadly reflects the pattern of lake number change illustrated in Figure 6c. In the Cordillera Real, there has been a 54 % increase in proglacial lake area from $2.7 \pm 0.27$ km$^2$ to $4.1 \pm 0.41$ km$^2$. The Tres Cruces region has seen a rather more modest increase of 15 % from $2.6 \pm 0.26$ km$^2$ to $2.9 \pm 0.29$ km$^2$, and the trend has levelled-off since 2010. Proglacial lakes in the Apolobamba region have seen a 51 % increase in area from $1.1 \pm 0.11$ km$^2$ to $1.7 \pm 0.17$ km$^2$, although the peak occurred in 2010 and lake area has since decreased. Figure 7b shows that total lake area has increased by 38 % from $6.33 \pm 0.63$ km$^2$ to $8.73 \pm 0.87$ km$^2$.

### 3.3 Identification of potentially dangerous lakes

In 2014, there were 137 lakes within 500 m of the 2014 ice margin, with a total area of $5.7 \pm 0.57$ km$^2$. From this database we identified 25 lakes that have the potential to be the source of damaging GLOFs (Table 2). Table 2 shows that these lakes are either moraine-dammed or sit within rock basins, and have surface areas that range across three orders of magnitude from 32,800 to 1,355,700 m$^2$. Complete drainage of the smallest lake would yield a flood with an estimated peak discharge of between 500 to 540 m$^3$s$^{-1}$. Complete drainage of the largest lake would yield an estimated peak discharge of 106,000 to 160,000 m$^3$s$^{-1}$, although the nine largest lakes are contained within rock basins and would be unlikely to drain completely. Many villages, farms and roads could be impacted by floods from the identified lakes. The lakes are also shown in Figures 3 to 5, although because of their size relative to the scale of the map, these are best viewed in the supplementary .kmz file.

## 4 Discussion

### 4.1 Bolivian glacier change

We make some comparisons with the limited previous research on Bolivian glacier change, although this is complicated to some extent because of inconsistent methodologies, different study periods, and inclusion or exclusion of glaciated areas in different inventories.

Glacier change in the Cordillera Apolobamba has not been investigated previously, which represents a significant gap in our understanding of Bolivian (and some Peruvian) glaciers. We have provided the first assessment of glacier change in this region. Jordan (1991, 1998) reported glacier areal coverage in this region for 1984 as 219.8 km$^2$. Our results from 1986 indicate ice coverage of $172.3 \pm 17.2$ km$^2$ for the Bolivian Cordillera Apolobamba, but this figure rises to $269.5 \pm 27.0$ km$^2$ when Peruvian glaciers are included (Figure 2b and 3). The discrepancy between our results and those of Jordan (1991, 1998) could be explained to some extent by further ice recession between 1984 and 1986. However, on closer inspection, it appears that Jordan (1991, 1998) included all ice across the Chaupi Orko range (see Figure 3 for location), both on the Bolivian and Peruvian sides of the border. When we include the same areas, the total for the Apolobamba Range is $221.3 \pm 22.1$ km$^2$, which is more consistent with Jordan's (1991, 1998) work. The slightly higher value is explained by our inclusion of some relatively small



glaciers separate from the main glaciated ranges of the Cordillera Apolobamba, which were not included in Jordan's (1991, 1998) mapping. Since 1986, the trend of glacier loss has been sustained throughout the study period, similar to other glaciated mountain ranges in Bolivia (Figures 2 to 5), with an overall glacier ice shrinkage of 43.1 % for the Bolivian Apolobamba, and 45.6 % for the combined Bolivian-Peruvian Cordillera Apolobamba.

Whilst individual glaciers of the Cordillera Real have been the subject of intensive study over many years (e.g. Ramirez et al., 2001; Sicart et al., 2011; Reveillet et al., 2015; Soruco et al., 2015), only the study of Soruco et al. (2009) has examined broader changes in glacier ice cover across these mountains. Their study demonstrated a 48 % surface area loss from 1975–2006, which is broadly consistent with our results, albeit for a different time window. The 1984 inventory presented in Jordan (1991, 1998) gives glacier ice cover of 323.6 km² for the Cordillera Real, which is consistent with our 1986 value of 315.2 ± 32.4 km²
(Figure 2c and 4).

Glacier change in the Tres Cruces region has been investigated by Albert et al. (2014) from 1975 to 2009. Their results indicated ~55 % areal loss over this study period, with a marked reduction in ice cover between 1975 and 1986, before the start of our monitoring period. Between 1986 and 2009, their results showed that ice cover had shrunk from ~36 km² to ~25 km², similar to our results (Figure 2d). Since then, our results have shown a further reduction of 12.4 % to ~22 km² for 2014.

Overall, the glacier retreat rate across the Cordillera Oriental is 1.54 % a⁻¹, excluding Peruvian glaciers of the Cordillera Apolobamba, or 1.57 % a⁻¹ if Peruvian glaciers are included. Regionally, the Tres Cruces experienced the highest retreat rates (1.69 % a⁻¹), followed by the Cordillera Apolobamba (1.54 % a⁻¹ excluding Peruvian glaciers, and 1.63 % a⁻¹ including Peruvian glaciers), with the Cordillera Real experiencing the lowest retreat rates (1.50 % a⁻¹). These values are comparable to retreat rates measured elsewhere in the Andes. For example, Rabatel et al. (2013) report retreat rates in Ecuador between 1962 to
1997 of 1.6 % a⁻¹, and 2 % a⁻¹ for Columbian glaciers from the late 1970s to early 2000s; retreat rates of between 0.34 % a⁻¹ and 2.05 % a⁻¹ are reported by Vaughan et al (2013) for glaciers in Peru across similar time periods. Retreat rates further south in Patagonia are reported to be much lower at 0.14 to 0.66 % a⁻¹ (Vaughan et al., 2013); in extra-tropical mountain ranges, such as the Alps, comparable rates of retreat of between 0.59 and 2.07 % a⁻¹ are reported, although retreat is generally reported to be lower than in the present study (Vaughan et al., 2013).

The trend in glacier shrinkage across the Cordillera Oriental is of some concern in terms of water resources across the region, and particularly for the major cities of La Paz and El Alto. These trends are likely to continue into the future. Although glacier shrinkage is not yet known to have driven rural to urban migration (Kaenzig, 2015), this could be a further pressure in the future, and it is likely that the effects of glacier shrinkage will be felt both in large cities, which rely to some extent on glacial meltwater (Vuille et al., 2008; Rangecroft et al., 2013; Soruco et al., 2015), and in rural communities (Andersen and Verner,
2009; Oxfam, 2009; Winters, 2012). Even in the Cordillera Apolobamba, which is rather sparsely populated, there are still ~5,500 people who live within 10 km of the glaciers (according to GeoBolivia GIS data - http://geo.gob.bo/). Approximately 13,700 people live within 10 km of the Tres Cruces glaciers, and ~30,000 people for the Cordillera Real. Future changes to glacial water supply are likely to be felt keenly within these immediate rural areas, where communities may depend to some extent on meltwater during the dry season for drinking water, crop irrigation, and sustaining livestock. However, glacial





meltwater also supplies populations in villages and cities beyond the immediate vicinity of the glaciated mountains. Another adverse impact would be toward the bofedales ecosystems (high altitude peat bogs and wetlands), which are also fed by glaciers, and which represent important water stores (Garcia et al., 2007; Squeo et al., 2009). Long-term glacier monitoring of the ice masses that supply water to La Paz and El Alto have been crucial in terms of understanding the sustainability of glacier

meltwater in the region (e.g. Reveillet et al., 2015), yet similar studies have not been undertaken in other mountain ranges across Bolivia (e.g. in the Cordillera Apolobamba and Tres Cruces) where local populations could be very vulnerable to future glacier shrinkage (Andersen and Verner, 2009; Oxfam, 2009; Winters, 2012).

To provide a first-order estimate of future glacier evolution across Bolivia, we used the data presented in Figure 2a to derive an exponential function (because it provided the best fit with the data compared to linear and other best-fit lines) that could be

used to model future glacier decay. This method indicates that glaciers across Bolivia will have shrunk to around 10 % of their 1986 area by ~2100. Extrapolation of glacier areal decline trends can only represent a first-order approximation, and masks the complex array of factors that determine glacier mass balance and volume, but our estimate suggests that further work is urgently required to accurately model glacier change, and to assess the consequences of that change on people and mountain ecosystems. There are few studies that model glacier demise in Bolivia, but Reveillet et al. (2015) modelled the future evolution

of Zongo glacier, forcing the model with temperature changes predicted by the Coupled Model Intercomparison Project phase 5 (CMIP5). Their results indicated that the Zongo glacier would lose 69 +/- 7 % of its volume by 2100 with the intermediate CMIP5 scenario, and 40 +/- 7 % and 89 +/- 4 % with the extreme scenarios. Although we are comparing an individual glacier with glacier demise across the whole Bolivian Andes, our results are consistent with the extreme upper end of these predictions.

### 4.2 Proglacial lake development

We have made the first evaluation of proglacial lake development across Bolivia. In general, the number of proglacial lakes and their areas (i.e. those that formed within 500 m of the 1986 ice margin) increased as glaciers have receded from their 1986 positions (Figures 6c, 6d, 7a, 7b). This shows that ice recession has revealed further basins that have filled with meltwater. Several studies have described similar trends of increasing number and size of proglacial lakes from a range of locations, both from ice-sheet and valley glacier contexts (e.g. Carrivick and Tweed, 2013; Carrivick and Quincey, 2014; Komori, 2008;

Loriaux and Casassa, 2013; Hanshaw and Bookhagen, 2014; Lopez-Moreno et al., 2014; Schomaker, 2010; Wang et al., 2014). Conversely, there has been an overall decrease in the number of ice-contact lakes across the study period for all three regions (Figures 6a and 7a). However, this trend has been very variable both spatially (across the three regions) and temporally. The trends in lake area have followed the changes in lake number for the Cordillera Apolobamba and Tres Cruces regions, but the Cordillera Real has experienced overall areal growth despite a reduction in lake number (Figure 6b). This is because there are

a few large ice-contact lakes in the Cordillera Real (e.g. Laguna Glaciar, Laguna Arkhata – Figure 4) that have been growing rapidly as ice has receded, in accordance with most findings in other deglaciating regions (e.g. Carrivick and Tweed, 2013; Komori, 2008; Loriaux and Casassa, 2013; Hanshaw and Bookhagen, 2014; Lopez-Moreno et al., 2014; Schomaker, 2010; Wang et al., 2014). Reductions in ice-contact lake number are explained by lakes becoming disconnected from the glacier. It





is nonetheless intriguing that lake number has generally decreased across all regions (Figure 6a and 7a), and lake area has decreased in the Cordillera Apolobamba and Tres Cruces regions (Figure 6b and 7b).

A few studies have also found that ice-contact lakes have reduced in number and/or size over time. For example, Gardelle et al. (2011) found that proglacial lakes in the Karakoram had shrunk because glaciers had surged or experienced reduced mass loss. This cannot explain the trends observed in Bolivia where there are no surge-type glaciers, and all glaciers are shrinking. Emmer et al. (2015) found that some lakes in western Austria had shrunk as a consequence of sedimentation and changes in water supply from the glacier. This explains the loss of some of the lakes within 500 m of the 1986 ice margin, but is less important for the evolution of ice-contact lakes. Apart from the documented GLOF case for the Apolobamba (Hoffmann and Wegenmann, 2013), we did not observe any further evidence for significant lake drainage events. Perhaps the simplest explanation is that, as the ice has receded, there is now a lower contact area between the ice and ice-marginal zone, and hence much less space within which ice-contact lakes can exist (the ice-contact perimeter reduced by 32.6 % between 1986 and 2014). Another possibility is that glaciers have now receded far behind their Holocene erosional maxima, where basins were carved out under thicker, faster-flowing ice (cf. Cook and Swift, 2012). Hence, as glaciers continue to recede, there are fewer deep basins being revealed that could provide accommodation space for ice-contact lakes to develop. Likewise, unlike the debris-covered glaciers of the Himalaya, which develop large terminal moraine complexes that enclose lakes as ice recedes (e.g. Hambrey et al., 2009), Bolivian glaciers are mostly clean-ice glaciers that do not generally develop large terminal moraines. Indeed, most of the potentially dangerous lakes are found in rock basins (Table 2).

Despite the recent trend of reducing ice-contact lake development, it should be emphasised that the trend in ice-contact lake number and area has been highly variable throughout the study period, indicating that future lake development could be unpredictable without further efforts to investigate proglacial lake appearance and evolution. New and large lakes could develop in the future. One promising avenue would be to measure or model glacier-bed topography in an effort to identify future locations of lakes (e.g. Frey et al., 2010), some of which could be dangerous with respect to GLOF risk.

### 4.3 Glacial lake outburst flood risk

An emerging issue in the Bolivian Andes is the threat of possible GLOF events (Hoffmann and Wegenmann, 2013). From our 2014 dataset of proglacial lakes, we undertook a first-pass assessment of lakes that could represent the greatest outburst flood hazard (Table 2; supplementary .kmz file), and hence should be the subject of future monitoring and GLOF modelling studies. A total of 25 lakes were identified that were large enough, and sufficiently close to potential sources of ice or rock avalanches, to be considered a potential GLOF risk to downstream communities or infrastructure.. Nine of the potentially dangerous lakes are moraine-dammed. Our estimations of peak discharge indicate potentially very damaging floods from all of the lakes identified in Table 2. Indeed, even relatively small lakes can generate damaging floods. Specifically, the ~34,000 $m^2$ ice-dammed lake that drained in 2009, damaging the village of Keara (Hoffmann and Wegenmann, 2013), produced a peak discharge of approximately 400 $m^3s^{-1}$ (a value that we calculated using supplementary ice-dammed lake data in Cook and Quincey, 2015, to derive an equation similar to Equation 3, and using Equation 4 to estimate peak discharge), and the erosional



and depositional evidence for the flood can be seen in GoogleEarth for ~10 km downstream. Hence, we recommend that future studies be directed toward more detailed modelling of potential floods from these lakes, more detailed hazard analysis taking into account the potential runout and inundation of the floods, as well as the vulnerability of the affected communities (e.g. Carey et al., 2014), and the monitoring of lake evolution and the development of new lakes. Bathymetric studies of the lakes

would also be welcome in order to improve volume estimations (Cook and Quincey, 2015).

## 5 Conclusions

Glaciers of the Bolivian Andes represent a crucial regional water source and hence there is significant concern with respect to the sustainability of that water supply in a changing climate. We have performed the first integrated study of glacier change across the Bolivian Andes. Our mapping from 1986 to 2014 revealed that there has been a reduction in glacier area from 529.3

$\pm$ 52.9 km$^2$ to 301.2 $\pm$ 30.1 km$^2$ across the study period, equivalent to a 43.1% shrinkage. Proportionally, ice loss was greatest in the southernmost part of our study area (Tres Cruces) where glaciers lost 47.3% of their area between 1986 and 2014. The Cordillera Real (middle part of our study area), represents the largest area of glaciation in Bolivia, and lost 41.9 % of its ice cover, while the Cordillera Apolobamba in the north lost 43.1% of its ice cover (or 45.6 % if the glaciers of the Peruvian Cordillera Apolobamba are also included). The trend in glacier recession has generally been rapid and continuous throughout

the study period.

We undertook the first assessment of proglacial lake development in Bolivia. There has been a general increase in the number and size of proglacial lakes across the study period, in accordance with several studies of proglacial lake development elsewhere. However, the trend in ice-contact lake number and area has been more complex throughout the study period. All regions show a net decrease in the number of ice-contact lakes through the study period, although this trend has been highly

variable. Whilst ice-contact lake area has experienced a net increase in the Cordillera Real, consistent with previous studies performed in other deglaciating mountain ranges, the area coverage in the Cordillera Apolobamba and Tres Cruces regions has decreased overall. It is unclear why these trends have emerged, but the variability in lake number and area indicates that ongoing monitoring of proglacial lake development is required, especially in light of the potential for these lakes to burst and initiate glacial lake outburst flood (GLOF) events.

GLOFs represent an emerging threat in Bolivia, with recent reports of GLOF events from the Apolobamba region (Hoffmann and Wegenmann, 2013). We used our 2014 (most recent) inventory of proglacial lakes to provide a first assessment of the potential for GLOFs in Bolivia. Overall, we identified 25 lakes that pose a potential GLOF risk to downstream communities or infrastructure. Estimated peak discharges from these lakes range from ~500 to ~160,000 m$^3$s$^{-1}$, although the upper end of these values could be unlikely given that the nine largest lakes are situated within rock basins, which are arguably more stable.

Nine of the lakes are moraine-dammed, and these could be susceptible to complete drainage. Sixteen lakes are rock-dammed, including the nine largest potentially dangerous lakes identified in this study. We recommend further monitoring of potentially dangerous lakes and modelling of GLOF hazards across Bolivia.





**Acknowledgements**

We thank the British Society for Geomorphology and the University of Manchester for research funding. IK is funded through an Environmental Science Research Centre studentship at Manchester Metropolitan University.

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





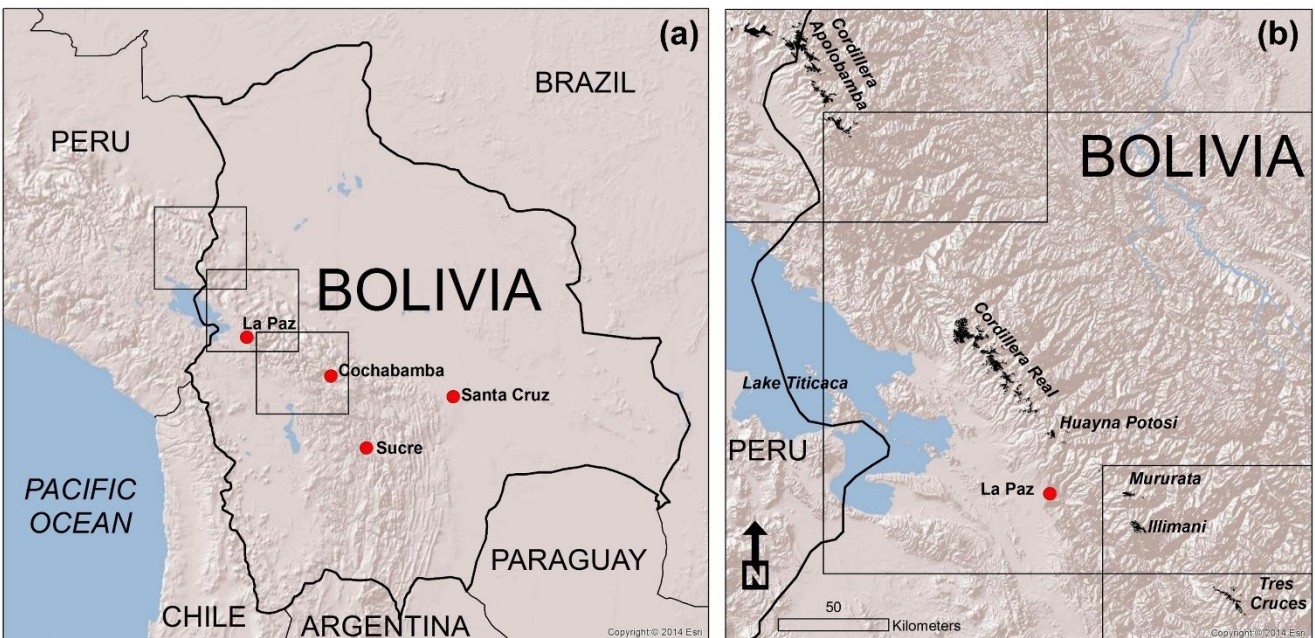

**Figure 1: Location of the study area. (a) Topographic map of Bolivia indicating the footprint (grey squares) of Landsat imagery used in this study; (b) Extent of the glaciated regions (shown in black as of 2014) of the Cordillera Oriental within the footprint of the satellite imagery. Base map is the Esri World Shaded Relief map, 2014.**




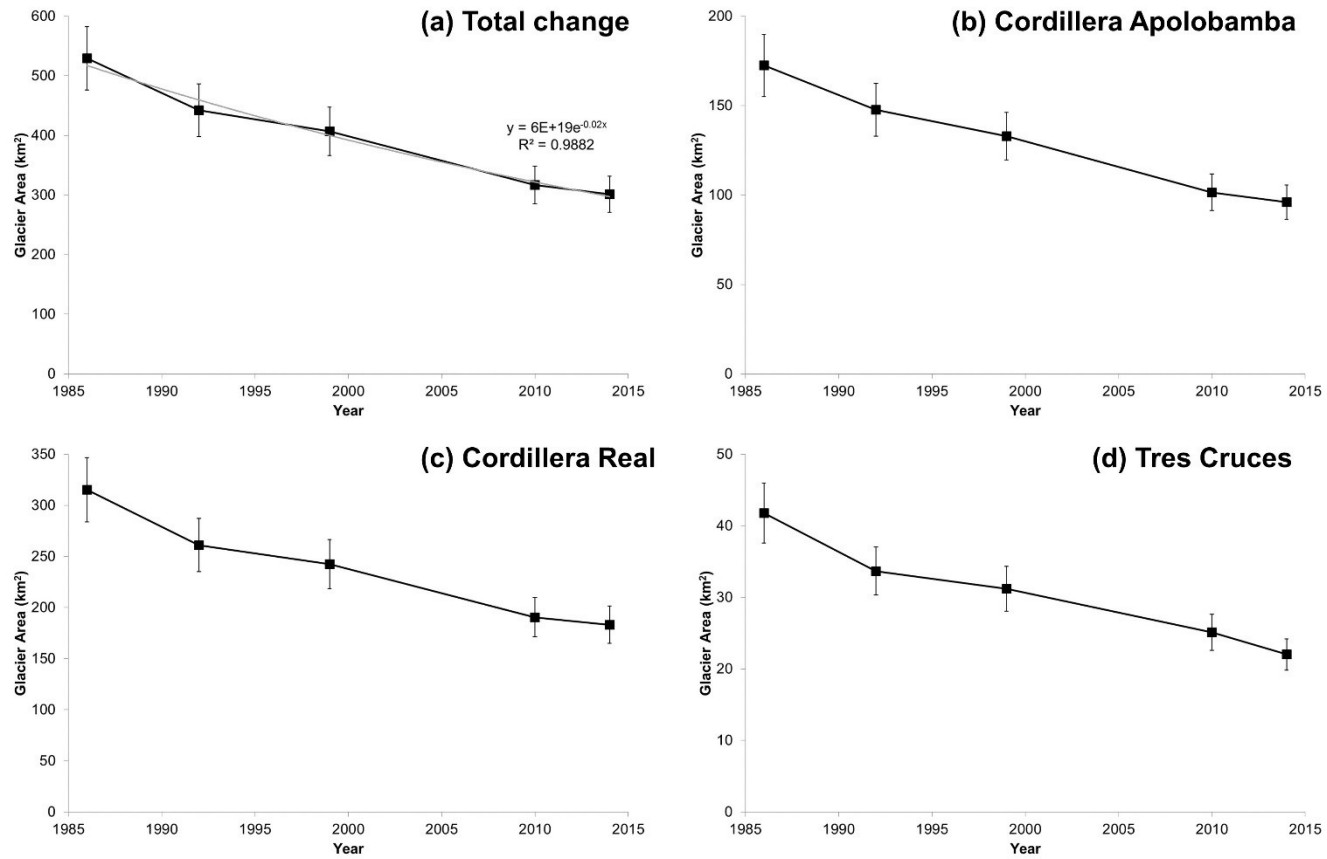

**Figure 2: Glacier areal change across the Bolivian Cordillera Oriental. (a) Total glacier area change (grey line is exponential best-fit relationship with associated equation and r² value); (b) Cordillera Apolobamba (excluding glaciers on the Peruvian side of the border); (c) Cordillera Real; (d) Tres Cruces region. Error bars are determined using Equation 1, and represent an uncertainty of ± 10 %.**



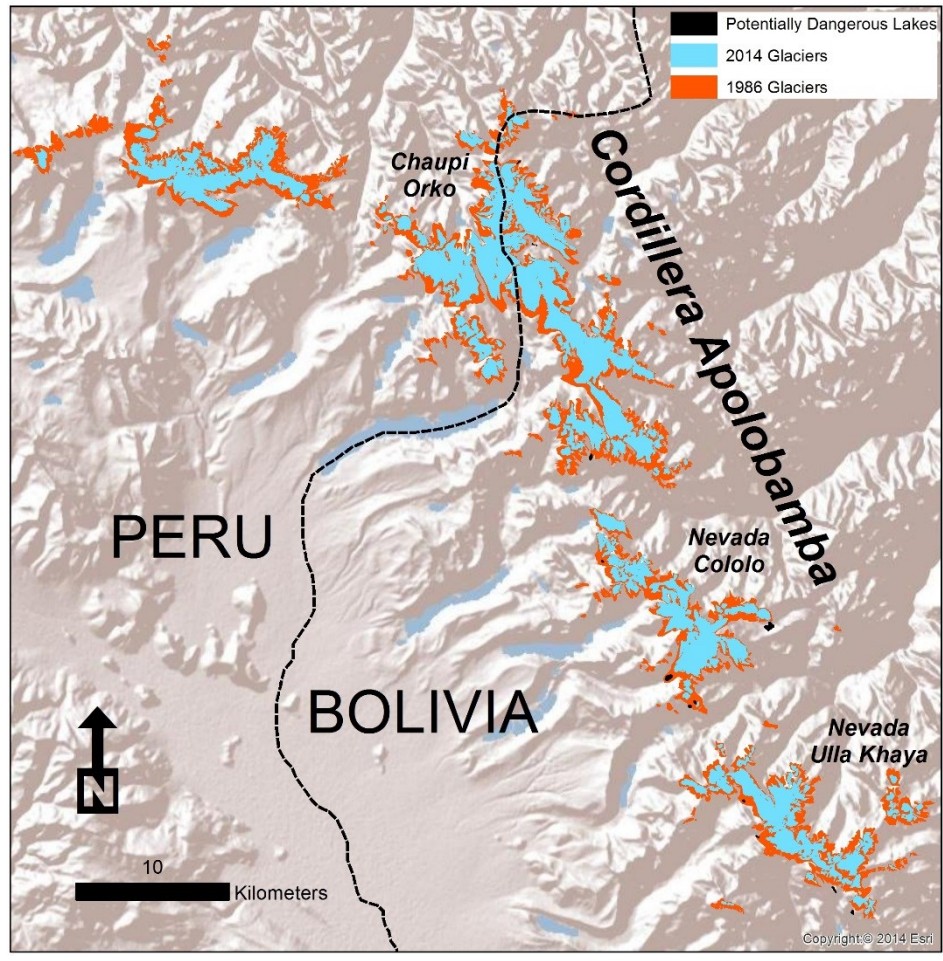

Figure 3: Glacier change 1986 to 2014 for the Cordillera Apolobamba. See Figure 1 for location of region. Base map is the Esri World Shaded Relief map, 2014.



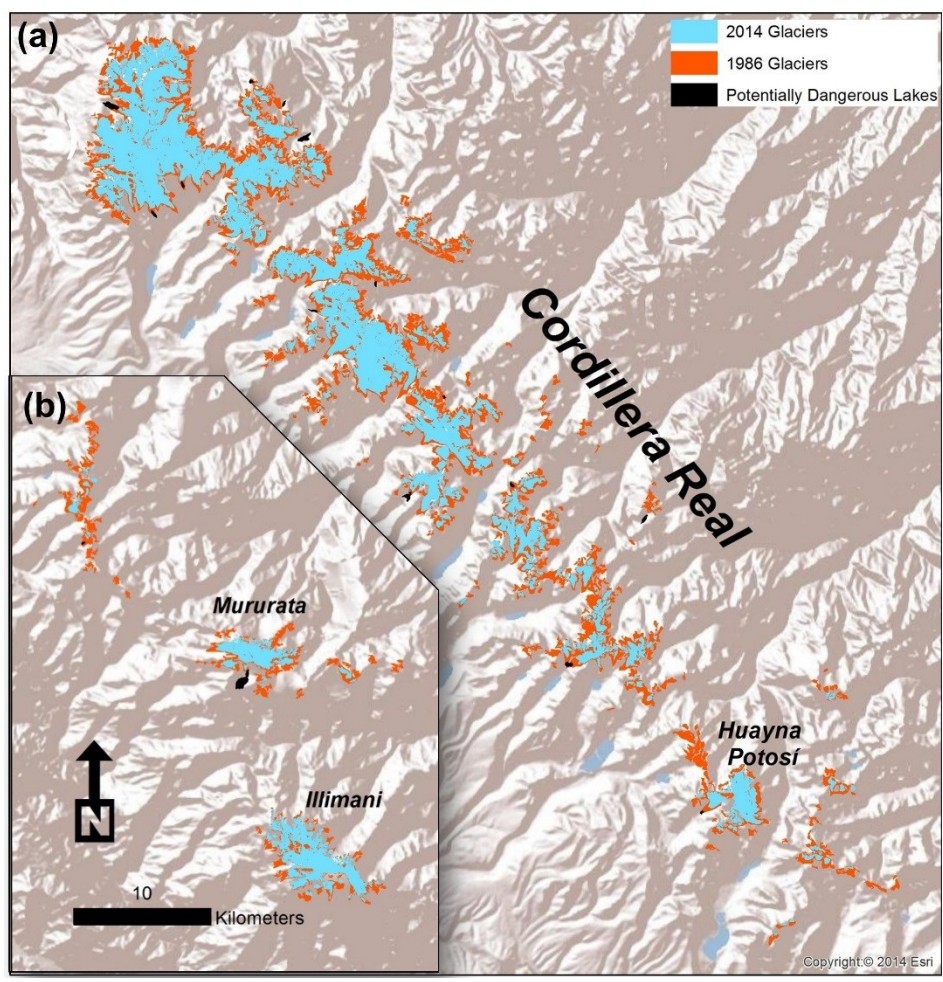

**Figure 4: Glacier change 1986 to 2014 for the Cordillera Real. Map (a) shows the main (northern) part of the Cordillera Real, and map (b) shows an inset of the southern Cordillera Real. See Figure 1 for location of region. Base map is the Esri World Shaded Relief map, 2014.**





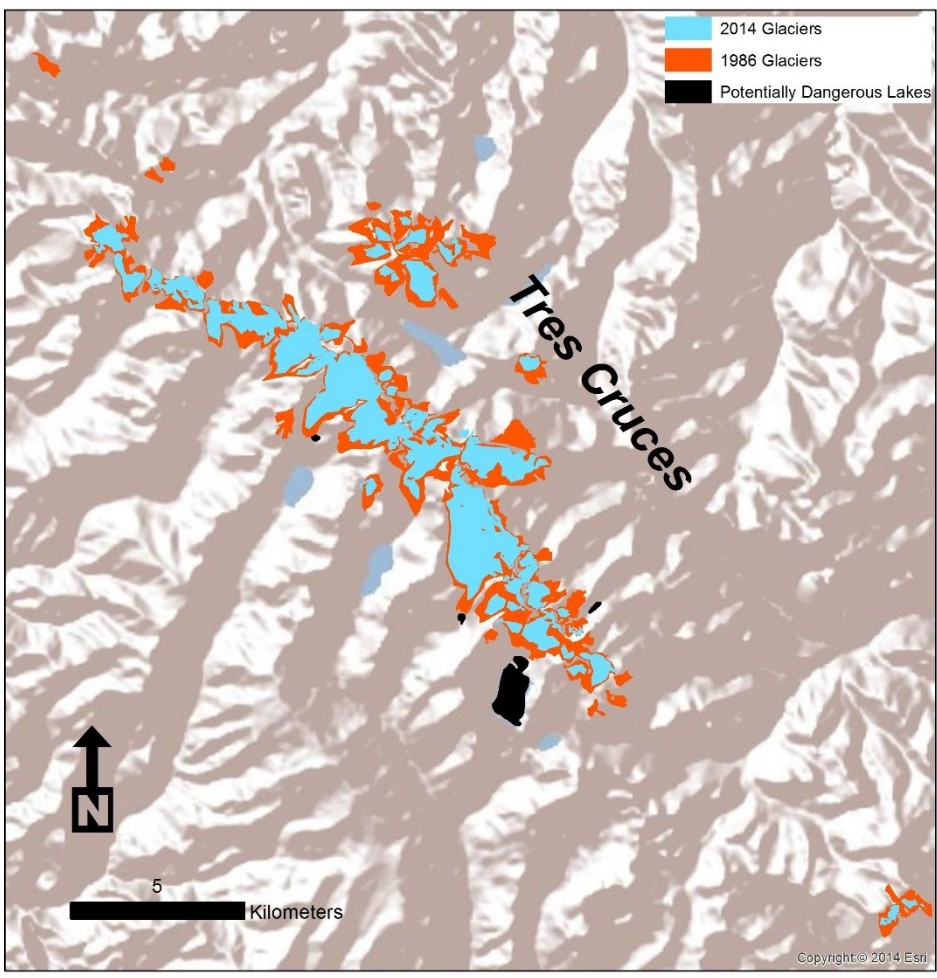

**Figure 5: Glacier change 1986 to 2014 for the Tres Cruces region. See Figure 1 for location of region. Base map is the Esri World Shaded Relief map, 2014.**



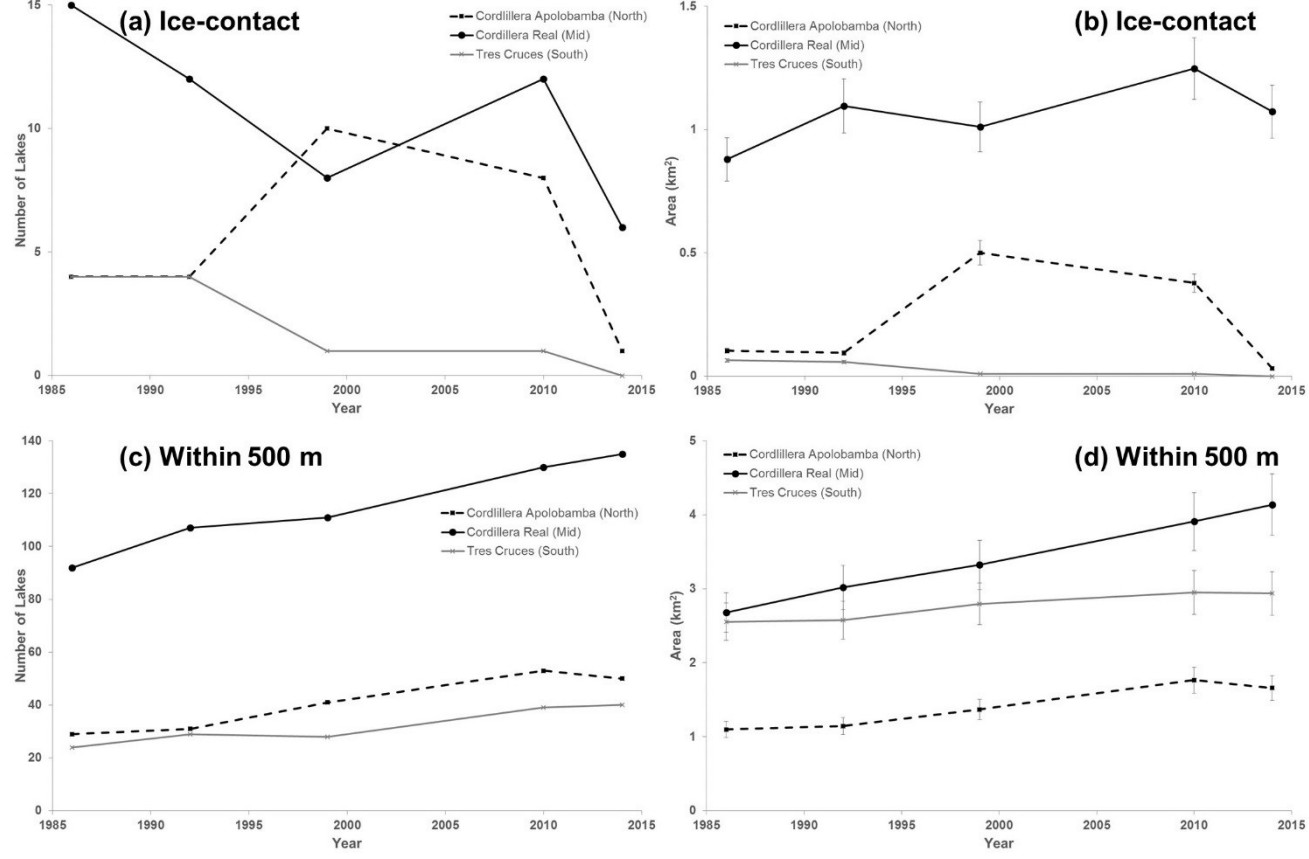

**Figure 6: Proglacial lake evolution from 1986 to 2014 across the Cordillera Apolobamba (North), Cordillera Real (Middle), and Tres Cruces (South) regions. The number and area of ice-contact lakes are shown in (a) and (b) respectively. The number and area of lakes within 500 m of the 1986 ice margin are shown in (c) and (d) respectively. Error bars are determined using Equation 1, and represent an uncertainty of ± 10 %.**



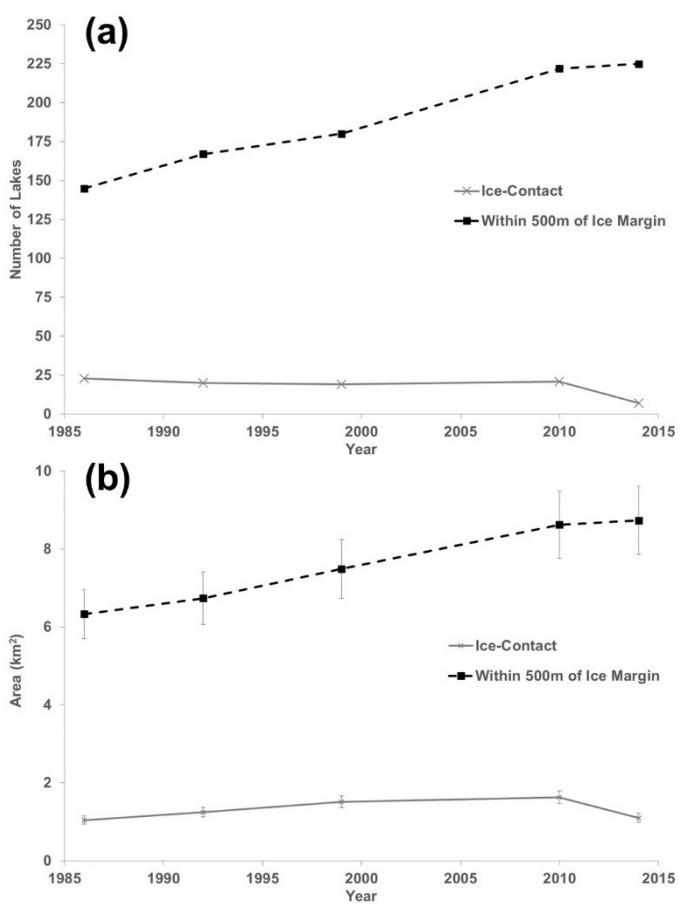

Figure 7: Total change in proglacial lake number (a) and areal cover (b) according to distance from ice margin (i.e. ice-contact, and within 500m of the 1986 ice margin). Error bars are determined using Equation 1, and represent an uncertainty of ± 10 %.



| Scene ID Number | Path/Row | Date | Sensor | Satellite | Cloud Cover (%) |
|---|---|---|---|---|---|
| LC80020702014215LGN00 | 002/070 | 03/08/2014 | OLI & TIRS | 8 | 8.81 |
| LC80010712014128LGN00 | 001/071 | 08/05/2014 | OLI & TIRS | 8 | 19.78 |
| LC82330722014153LGN00 | 233/072 | 02/06/2014 | OLI & TIRS | 8 | 8.62 |
| LT50020702010220CUB00 | 002/070 | 08/08/2010 | TM | 5 | 0 |
| LT50010712010261CUB00 | 001/071 | 18/09/2010 | TM | 5 | 7 |
| LT52330722010126CUB00 | 233/072 | 06/05/2010 | TM | 5 | 0 |
| LT50020701999222COA03 | 002/070 | 10/08/1999 | TM | 5 | 7 |
| LE70010711999255COA01 | 001/071 | 12/09/1999 | ETM+ | 7 | 18 |
| LT52330721999224COA03 | 233/072 | 12/08/1999 | TM | 5 | 0.14 |
| LT50020701992203CUB00 | 002/070 | 21/07/1992 | TM | 5 | 10 |
| LT50010711992212CUB00 | 001/071 | 30/07/1992 | TM | 5 | 21.91 |
| LT52330721992173CUB00 | 233/072 | 21/06/1992 | TM | 5 | 7 |
| LT50020701986298XXX03 | 002/070 | 25/10/1986 | TM | 5 | 10 |
| LT50010711986227XXX04 | 001/071 | 15/08/1986 | TM | 5 | 30 |
| LT52330721986220CUB03 | 233/072 | 08/08/1986 | TM | 5 | 10 |

**Table 1: Summary of Landsat scenes used in this study.**





| Location | Coordinates | Lake Type | Area (m²) | Volume (m³) Equation 1 Huggel et al. (2002) | Volume (m³) Equation 2 Cook & Quincey (2015) | $Q_{max}$ (m³ s⁻¹) Equation 3 Huggel et al. (2002) | $Q_{max}$ (m³ s⁻¹) Equation 3 Cook & Quincey (2015) | Potential Effect |
|---|---|---|---|---|---|---|---|---|
| Apolobamba – Puina (1) | 476504, 8384832 | Moraine-dammed | 32800 | 269000 | 248000 | 540 | 500 | Isolated farms; Village in Puina district; damage to road |
| Apolobamba - Taypi Cayuma (2) | 491182, 8343142 | Rock basin | 34900 | 293000 | 273000 | 590 | 550 | Village of Taypi Cayuma; damage to road |
| Apolobamba - Taypi Cayuma (3) | 492072, 8340807 | Moraine-dammed | 35500 | 301000 | 280000 | 600 | 560 | Village of Taypi Cayuma; damage to road |
| Tres Cruces (4) | 670245, 8126070 | Moraine-dammed | 40500 | 363000 | 344000 | 730 | 690 | Damage to road |
| Apolobamba - Hilo Hilo (5) | 487996, 8349572 | Rock basin | 45600 | 429000 | 413000 | 860 | 830 | Villages of Hilo Hilo; damage to road |
| Cordillera Real - Comunidad Pantini (6) | 612872, 8182149 | Rock basin | 48400 | 467000 | 453000 | 930 | 910 | Isolated farms; bridges and road |
| Apolobamba - Hilo Hilo (7) | 487666, 8349316 | Rock basin | 48500 | 468000 | 455000 | 940 | 910 | Villages of Hilo Hilo; damage to road |
| Tres Cruces (8) | 674446, 8120893 | Moraine-dammed | 53600 | 540000 | 532000 | 1080 | 1060 | Mining camp; damage to road |
| Apolobamba - Cholina Cholina (9) | 498284, 8335884 | Moraine-dammed | 54300 | 550000 | 542000 | 1100 | 1080 | Damage to road; flooding of agricultural fields |
| Tres Cruces (10) | 678278, 8121207 | Rock basin | 62400 | 670000 | 673000 | 1340 | 1350 | Isolated homesteads; damage to road |
| Cordillera Real – Cocoyo (11) | 556846, 8251418 | Rock basin | 66700 | 737000 | 747000 | 1470 | 1490 | Village of Cocoyo; damage to road |
| Cordillera Real – Cocoyo (12) | 559120, 8249880 | Rock basin | 68300 | 761000 | 774000 | 1520 | 1550 | Village of Cocoyo; damage to road |
| Apolobamba - Cholina Cholina (13) | 497085, 8337363 | Moraine-dammed | 70900 | 803000 | 820000 | 1610 | 1640 | Damage to road and isolated farms |
| Cordillera Real – Rinconada (14) | 552071, 8244232 | Moraine-dammed | 76000 | 887000 | 915000 | 1770 | 1830 | Damage to road |
| Apolobamba – Pelechuco (15) | 481205, 8365591 | Moraine-dammed | 80000 | 954000 | 990000 | 1910 | 1980 | Direct damage in Agua Blanca and Pelechuco; damage to road |
| Cordillera Real – Rinconada (16) | 550069, 8242190 | Moraine-dammed | 102700 | 1360000 | 1460000 | 2720 | 2920 | Damage to road |
| Cordillera Real – Umapalca (17) | 584186, 8220965 | Rock basin | 130500 | 1910000 | 2120000 | 3820 | 4230 | Village of Umapalca; damage to road |
| Cordillera Real – Condoriri (18) | 578927, 8210860 | Rock basin | 146800 | 2260000 | 2540000 | 4520 | 5080 | Isolated homesteads; damage to road |
| Apolobamba - Puyo Puyo (19) | 486275, 8351196 | Rock basin | 154800 | 2440000 | 2760000 | 4870 | 5520 | Damage to road |
| Cordillera Real - Laguna Wara Warani (20) | 567694, 8222503 | Rock basin | 201800 | 3550000 | 4165000 | 7100 | 8330 | Village of Halluaya; damage to road |
| Apolobamba - Hilo Hilo (21) | 492850, 8354529 | Rock basin | 254500 | 4932000 | 5969000 | 9860 | 11900 | Isolated farmsteads and dwellings of Chiata community damage to road |
| Cordillera Real – Cocoyo (22) | 560553, 8247486 | Rock basin | 289400 | 5920000 | 7288000 | 11800 | 14600 | Village of Cocoyo; damage to road |





| | | | | | | | | |
|---|---|---|---|---|---|---|---|---|
| Cordillera Real - Laguna Glaciar (23) | 547085, 8249728 | Rock basin | 328600 | 7090000 | 8877000 | 14200 | 17800 | Damage to roads, agricultural land, several villages; popular tourist destination |
| Laguna Arkhata, Mururata (24) | 624521, 8172040 | Rock basin | 699200 | 20719000 | 28676000 | 41400 | 57400 | Village of Totorapampa and Tres Rios; damage to road |
| Tres Cruces - Laguna Huallatani (25) | 675910, 8118767 | Rock basin | 1355700 | 53055000 | 80180000 | 106000 | 160000 | Isolated homesteads; damage to road and agricultural land |

**Table 2: Compilation of potentially dangerous lakes across the Bolivian Andes. Area, volume and discharge values given to 3 significant figures. Number in brackets in 'Location' column refers to lake identification number in Supplementary .kmz file.**

