# Peer review of "Glacier change and glacial lake outburst flood risk in the Bolivian Andes"

_The Cryosphere, 2016_

## Short Comment (SC1) · 18 Jul 2016

Cook et al: 2016 provide a detailed examination of glacier area change and glacier lakes in key glaciated regions of Bolivia. This is a valuable paper that advances both regional knowledge of glacier and glacier lake change, but also utilizes efficient and reliable techniques. The GLOF threat and the current trend in that threat may be overstated, given the current decline in ice-contact glacier lakes noted and the author's observation that few new lakes seem to be forming and the lakes will likely have reduced sizes. The impact on glacier runoff of the 43% glacier area loss is likely understated and should be better quantified, though still being a first order approximation.

Specific Comments: 2-27: The lack of attention is likely due to their not being any historic experience with GLOF suggesting the risk is not high in this region. Hoffman

and Weggenmann (2012) point out that the Keara example here is the first known GLOF in the area.

4-1: The step wise approach for glacier area identification is a best practice approach.

4-27: If these lakes are not in contact with the glacier and the steep slope has not been recently deglaciated, than is GLOF is the correct term?

5-1: I assume this applies to only recently deglaciated steep slopes, if not just state.

5-28: Reword; Lakes confined within rock basins are less likely to experience breach incision.

6-3: Any examples of application of this formula in the Andes?

7-10: This is one of the key finding that all regions had a decline in ice contact lakes from 1986 to 2014, Figure 6a does not communicate this as well as a Table would.

7-17: A Table would provide a better display of lake number changes than Figure 6a or 6b, since these are not actual trends through the study period, a line chart display does not provide a clear picture of what is occurring. A table could also better quantify the total rock basin versus moraine dammed lakes.

8-10: Were the number of dangerous lakes determine at any other time than 2014? If so how many were there? How many of these 25 are ice-contact?

9-26: Can emphasize this point with some quantification. A 43% decline in glacier area suggests that total glacier runoff would have already declined substantially as runoff is product of glacier area and ablation rate. This will also affect timing as noted by below studies. "Despite a 15% increase in ablation rate, the 45% decline in glacier area led to a 38% decrease in glacier runoff in the Skykomish River basin" (Pelto, 2011). For specifics on the tropical Andes and seasonal impacts, Vuille et al. (2008) is useful. Some areas with losses of 20% of glaciated area have documented declines in glacier runoff and timing changes, that will be occurring here as well Stahl and Moore (2006),

Dery et al. (2009) and Pelto (2011).

9-29: Is glacier water supply change felt keenly? This is a region without significant hydropower or fisheries; hence the impact would be on municipal and agriculture impacts. This is an impact with much wider potential importance than GLOF's and deserves further attention. Provide some information on annual precipitation in the region beyond the glaciers and some understanding of the relative water supply from glaciers to this area. A sentence or two defending the statement would prove valuable.

10-26: The decline in ice-contact lakes suggests the number of pro-glacial lakes will decline as glacier retreat continues in the near future. Is this true?

11-7 to 11-14: This discussion illustrates that GLOF risk should be declining as the authors note fewer new lake basins are being exposed or likely to be exposed and their size is reduced. This should be emphasized in abstract and conclusion.

12-5: Is there any chance in your view that remote sensing could be used to assess lake depth in combination with some ground truth? If so elaborate, if not leave alone.

12-25: Given the statements on 2-30 indicating a lack of previous damaging GLOF's and the decline in ice-contact lakes, I do not see how emerging can be used. If anything the data here suggests GLOF's risk will decline. It is certainly appropriate to emphasize the prior lack of quantification of the risk.

Figure 3: Is the Bolivia Peru Border in the Chaupi Orko Massif too far east?

Dery, S. Stahl. K. Moore, R. Whitfield, W. Menounos, B. and Burford J.: Detection of runoff timing changes in pluvial, nival and glacial rivers of western Canada. Water Res. Res. 45: doi:1029/2008WR006975, 2009.

Hoffmann, D. and Weggenmann, D.: Climate Change Induced Glacier Retreat and Risk Management: Glacial Lake Outburst Floods (GLOFs) in the Apolobamba Mountain Range, Bolivia. In: Climate change and disaster risk management, Springer,2013.

Pelto , M.: Skykomish River, Washington: Impact of ongoing glacier retreat on stream-flow. Hydrological Processes 25(21): 3267–3371, 2011.

Stahl, K and Moore, R.: Influence of watershed glacier coverage on summer streamflow in British Columbia, Canada. Water Resources Res. 42: doi: 10.1029/2006WR005022, 2006.

Vuille, M., Francou, B., Wagnon, P., Juen, I., Kaser, G., Mark, B. G., and Bradley, R. S.: Climate change and tropical Andean glaciers: Past, present and future, Earth-science reviews, 89, 79-96, 2008.

---

## Referee Comment (RC1) · W. Haeberli (Referee) · 27 Jul 2016

General

In agreement with the editorial comments by Etienne Berthier and the comments by Mauri Pelto I consider this contribution to be a welcome and interesting study about a less well-documented regional development. As the two mentioned colleagues discussed the questions of remote sensing, glacier mapping and hydrology, my comments can focus on questions of hazard and risk assessments. My recommendations aim at encouraging the authors to more critically reflect their techniques and formulations concerning people-related hazards and risks. Scientific hazard and risk studies are policy-relevant and require aspects of transparency and honesty to be especially critically reflected.

[Figure]

Hazard and risk aspects

Equations (2) and (3) are used for area-based estimates of lake volumes, which are then applied as input for calculating peak-discharge values, which in turn are taken as indicators of hazard potentials. There are three fundamental problems related to the application of such unfortunately quite popular equations: Volume-area relations are unnecessary area-area self-relations, the preciseness of the numerical values used in the regression equations is disproportionate with respect to the accuracy and reliability of the results obtained and the statistics deal with mean instead of extreme values which is an uncommon and problematic procedure in hazard consideration.

Lake volumes are determined by multiplying measured lake areas with measured and averaged/integrated lake depths. Correlating lake volume with lake area, therefore, means to correlate a mathematical product with one of the factors from which it had been calculated. Correspondingly, predicting lake volumes from lake areas means to essentially predict lake areas as used for volume calculation from themselves. Why should we do this? Because "many do it" (including myself in earlier papers)? Because the statistics and scatter plots look better? Or because the knowledge on how lake volumes are determined is lost? I strongly recommend to strictly avoid unnecessary volume-area self-relations but to use the straightforward relation between the originally determined lake areas and lake depths. This straightforward approach produces exactly the same results but is transparent and honest in that it not only shows what is measured and what is calculated but especially also illustrates the large scatter in the relation between the two measured variables and shows the resulting enormous uncertainty in the estimated values as well as in all further values (here volumes, peak discharges) derived from them. In fact, the morphometric analysis of glacier-bed overdeepenings where lakes may form clearly shows that large features can be shallow and small features can be deep (Haeberli et al. 2016). This should be clearly said in order to avoid unrealistic expectations: Orders of magnitude for lake volumes can at best be estimated empirically. Correspondingly, the excess preciseness of the numerical values used in equations (2) and (3) derived from statistical regression and used in such order-of-magnitude estimates is disproportionate and provides a misleading impression of the accuracy and reliability which can be reached in reality. In addition to these two problems, equations (2) and (3) also have the problem that they represent medium-value statistics while hazard assessment must be made with extreme-value statistics for worst-case considerations.

Equation (4) is an example and outcome of such reflections: It avoids over-sophisticated and over-precise mean-value and self-relation statistics but enables in a perfectly transparent way the realistic estimation of empirical extremes in a simple way even without any computer: the fact that the right-hand side of this equation is written as 2V/1000 instead of V/500 helps to make its application as easy as possible, even in the field or for non-scientists. This simplicity and transparency also makes it clear to scientists as well as to stakeholders or even the public where the limits are of our knowledge, understanding and ability to predict numbers of unmeasured lake volumes for practical applications in the real world.

Extreme peak-discharge values as estimated, for instance, by equation (4) refer to worst-case events. In the case of glacial and periglacial lakes, such extreme peak values can result from sudden-break mechanisms of dams consisting of broken ice from ice avalanches or glacier surges, from massive erosion and debris-flow formation in connection with moraine breaching, or from squeezing-out of more or less entire lakes by large ice/rock avalanches. This is again essential to be made clear. Worst-case scenarios with such extreme peak discharge values relate to low-frequency/high-magnitude events. Dealing with corresponding hazards from events with extremely low probability but also with extreme damage potential are a special challenge for policy making with regard to risk acceptance and risk management. Most outburst processes such as, for instance, progressive enlargement of sub-glacial channels produce far smaller peak discharges. Careful wording of worst-case scenarios is necessary in order to avoid adverse psychological and economic effects which can exceed the po-

tential damage occurring in reality or perhaps even not occurring at all.

The continued glacier retreat indeed tends to reduce glacier sizes and, hence, areas and volumes of new lakes forming as a consequence of ice vanishing. It also reduces direct ice-contacts of lake water with calving fronts. This does, however, not necessarily reduce the hazard potential. The more the glaciers retreat the closer new lakes form to steep walls of icy peaks with degrading permafrost and long-term reduction of slope stability. The corresponding scientific literature is easily accessible today. In Bolivia, permafrost can be expected at altitudes above about 5000m a.s.l where the 0°C isotherm is found (cf. Carey et al. 2012). This may be a lesser influence on the hazard situation in the investigated region but must be correctly mentioned and treated. De-buttressed slopes and slopes with degrading permafrost are the two situations with the most rapid and important change in long-term rockfall disposition.

Minor technical notes

1-24: "contain" not contains

2-04: The term "climate warming" is popular but not scientifically correct and should better be replaced by something like "climate change", "global warming" or "atmospheric temperature increase": Climate is defined as an average of meteorological parameters (not only temperature) over extended time periods (usually decades). As such it cannot "warm" (the permafrost can warm).

2-26: Carey et al. (2012) and Haeberli et al. (2016; as mentioned in the text) could be added here concerning the recent example of Laguna 513 in the Cordillera Blanca).

4-15 and 5-first paragraph: The 500m limit is highly subjective and not really reliable: Ice avalanches over firn/ice surfaces can have trajectory lengths 3 times the drop height and rock or rock/ice avalanches can by far exceed such limits. If the authors prefer to stick to their number they should comment on it accordingly. The phenomenon of permafrost should be mentioned here.
5-9: The involved processes of outburst triggering and flood propagation in the torrent below the lakes are of primary importance concerning potential damages and corresponding hazard potentials, rather than lake area or lake volume. This should be mentioned.

5-13/14: This assumption is again delicate: squeezing out of a small lake by an ice or rock avalanche may cause far-reaching floods with high damage potential.

5-23: Trajectory slopes have so far been used rather than simple distances. Process chains can affect infrastructure over much longer distances than 20 km. Again: A problem of low frequency/high magnitude events. Most events will remain far within distances of 20 km but in the worst case 20 km may by far not be enough.

7-33: Eliminate "increase" after 72%.

8-10/14: make clear that these values refer to highly unprobable worst-case scenarios.

11-01: ". . . decreased in such cases . . ."?

11-23: a more recent reference would be Linsbauer et al. (2016)

11-28: eliminate one full stop after ". . . infrastructure."

12-28/29: mention that these values are worst-case scenarios and add that the "higher stability" of lakes in rock basins may be questioned in case of possible impact waves produced by rock/ice avalanches from de-buttressed slopes or slopes with degrading permafrost (cf. especially Deline et al. 2014).

Table 2: Lake volumes are not extreme values but peak discharges are extreme values from worst-case scenarios. At least point to this discrepancy and discuss it in the text

References

Carey, M., Huggel, C., Bury, J., Portocarrero, C. and Haeberli, W. (2012): An integrated socio-environmental framework for glacier hazard management and climate change

adaptation: lessons from Lake 513, Cordillera Blanca, Peru. Climatic Change112, 3, 733-767.

Deline, P., Gruber, S., Delaloye, R., Fischer, L., Geertsema, M., Giardino, M., Hasler, A., Kirkbride, M., Krautblatter, M., Magnin, F., McColl, S., Ravanel, L., Schoeneich, P. (2014): Ice loss and slope stability in high-mountain regions. In: Haeberli, W., Whiteman, C. (Eds.), Snow and Ice-related Hazards, Risks and Disasters. Elsevier, Amsterdam, 521–561.

Haeberli, W., Linsbauer, A., Cochachin, A., Salazar, C. and Fischer, U.H. (2016): On the morphological characteristics of overdeepenings in high-mountain glacier beds. Earth Surface Processes and Landforms. doi:10.1002/esp.3966

Linsbauer, A., Frey, H., Haeberli, W., Machguth. H., Azam, M.F., Allen, S. (2016): Modelling glacier-bed overdeepenings and possible future lakes for the glaciers in the Himalaya–Karakoram region. Annals of Glaciology 57 (71), 119-130. doi:10.3189/2016AoG71A627

---

## Referee Comment (RC2) · Anonymous Referee #2 · 2 Sep 2016

General comments:

This paper fills a gap identified in the literature for an updated inventory of glacier mass changes and formation of potentially dangerous lakes in Bolivia. There are relatively few glaciers and a total glacier lake area of less than 10 km2 in Bolivia. Still, apparently there has not been a comprehensive and up-to-date accounting of glacier and lake changes for all ranges assessed here. This paper nicely provides a time trajectory of change over the most recent decades, when it has been widely recognized that change is occurring at unprecedented rates. The observations are therefore inherently valuable and should be published.

The paper is well-written and clear, and the methods of acquisition sound. However, the scientific merit of the data analyses is less obvious in terms of advancing process

understanding from the basic observations (i.e. this is clearly a reporting of important observations; what are the implications in terms of our understanding of dangerous lake formation?).

Specific comments:

Care should be taken in how related issues of water resources and vulnerability are included. It should be clarified that without measured values, the discussion of water supply and urban migration remains speculative. For example, what is the basis for estimating 15% of potable water to El Alto and La Paz come from glaciers (P1-28)? And is this water originating from a net loss of glacier mass, or is it simply from glacier fed streams? Without specifying, these general statements amount to speculative hyperbole. The authors should provide numbers, and references. Similarly, much of La Paz power comes from hydroelectrity? How much power generation is really vulnerable to being unreliable with low flows?

There is a sequence of steps taken to both automatically analyze the scenes, and also use human expertise to evaluate "dangerous lakes" vis-a-vis factors of risk and damage potential. Why not present this algorithm as a flow diagram? And why not provide relative scale/magnitude of danger? It is interesting to consider the area of lakes that meet the threshold of 'dangerous'. Why not present a graduated ranking of significance from least to most dangerous? The inventory of lakes seems to be numbered from smallest to largest volume/Qmax. But even that is not explicit in the text.

Error is equated to uncertainty, determined to be 10%. Is this rather high? This is based on an assumption of Gaussian distribution. I'm just unclear about how this relates to the reporting of areas.

I do like the inclusion of a kmz file for viewing the lakes. Simple and effective.

In assessing the lake formation related to glacier change, it seems that more analyses

of basic variables could be easily explored to substantiate some of the patterns that the authors observe, and help make more robust suggestions about processes. I think it most compelling to explore pattern of lake changes that might be anticipated given the valley morphology. For example, how do patterns of lake changes relate to topographic indices (e.g. hypsometry, lake elevation, distance from headwall)? What about relating lake changes to glacier forms? These seems straight forward derivatives of this impressive database that has been generated. Then, it would seem the authors might articulate more clear hypotheses to explain the lack of trend in ice-contact lake formation over time. How do these patterns compare to other regions?

It is not clear: were moraine-dammed lakes considered more dangerous or not? If there is no difference, why bother categorizing them as such? Explain.

I wonder why the lakes are numbered as they are (by size of Q max, starting from smallest?). Why not by region? Or keeping #1 as the largest (i.e. reflecting risk magnitude)?

Throughout this discussion, it is probably misleading to talk about "trend" with respect to the change in ice-contact lakes since there is not a significant tendency to the data.

Technical corrections:

P3 L5: what is meant by "first-pass assessment"? Is it spelled with dash or not (compare P4L23)?

P4L26 (and elsewhere): use colon before numbered list.

P5 L23 "Direct hydrological connection" to downstream infrastructure and communities

P5 L27: Lakes "sat" should be set

P8 L21: use 'glacierized' for currently glacier covered, as glaciated can refer to previously ice covered

P10 L27: "very variable" is awkward phrase; better to use "highly variable"
P11 L28: double periods.

P11, L32,33: remove the parentheses as this is a substantive point.

Map figures: the ESRI hillshade backdrop is okay, but they lack of any downstream features (population centers, roads) that are used in evaluating the 'dangerous' lake conditions.

Table 2: separate lake number and list by range. Could also abbreviate lake type to save space.

[Figure]

---

## Author Comment (AC1) · 15 Sep 2016

We thank Mauri Pelto for his insightful comments and questions about our manuscript. This provides us with an opportunity to elaborate on some points in our study. Taking each point in turn:

2-27: It's hard to quantify exactly what the level of GLOF risk is in this region. One of the points of our paper is to highlight that there is a risk, and that this risk has not been assessed before. Hoffmann & Wegenmann (2013) described the only documented GLOF in Bolivia, as far as we are aware, but that does not necessarily mean that it is the only GLOF that has occurred here. It is worth remembering that, for the most part, the people affected by such events live in rather remote communities. GLOF events may simply go undocumented. Last year, we travelled to Agua Blanca, in the

Apolobamba Region (Northern part of our study area), where we spoke to the village leader about the work we were doing there and this led him to recount a story from his father who had witnessed a GLOF event some years ago in a neighbouring valley. There is no documentation of this event, and it is hard to verify, but the point is clear: there may be other undocumented GLOFs that have taken place in Bolivia.

4-1: OK

4-27: I think this is a reasonable point. I suppose the issue is whether a lake that occupies a basin that has been carved out by a glacier or dammed by glacial sediment constitutes a 'glacial lake', and hence if it were to burst can be considered a 'glacial lake outburst flood'. I agree that the term 'GLOF' would normally be used for outbursts from ice-contact lakes or lakes that had formed 'recently' following deglaciation (whatever you take 'recent' to mean). I hope that most readers would appreciate what is meant when we use the term 'GLOF' in the context of our work in Bolivia.

5-1: Not necessarily. In Bolch et al.'s (2008) paper, the whole study region is classified according to slope, with the slopes of 45 deg or steeper representing the greatest mass movement risk. They aren't necessarily recently deglaciated. We just wanted to make the extra point here that recently deglaciated slopes can be particularly susceptible to mass movements.

5-28: Good idea. We have reworded.

6-3: No, not that I am aware of. In compiling data for an earlier study (Cook and Quincey, 2015), we generated a list of studies that had employed the lake volume-area relationship of Huggel et al. (2002). At the time of doing that (c. 2014-15) there were some studies that borrowed the relationship to estimate lake volume for glacial lakes in the high mountains of Asia (Nepal, Tien Shan, etc), but none for the Andes. We have subsequently removed the equation from Huggel et al. (2002) in response to comments by Reviewer 1 (Wilfried Haeberli).

7-10: I suppose the extent to which a table vs. a figure best illustrates a trend in data is debateable. But I would like to emphasise a key issue here – we are not only making the point that ice-contact lakes have declined over time, but also that the pattern of change over time has been rather chaotic. I would argue that this is best illustrated with a figure because it is more immediately comprehensible.

7-17: As with the previous point, I would argue instead that the graph shows the complexities in lake change over time more effectively than a table of numbers, which would be harder to digest. I take the point that a Table would be useful in terms of information on lake type, but this is perhaps most relevant for potentially dangerous lakes – for these lakes, information on moraine-, rock-dammed lakes, etc, is provided in Table 2.

8-10: No, we did not determine the number of dangerous lakes for other years. Our reasoning for this is that, although understanding how the GLOF risk has evolved over time would certainly be interesting, the more pressing issue is to determine which of the lakes from the most recent dataset would be most likely to represent a GLOF risk. This seemed to us to be the more urgent question, and hence the one that should be given over to journal page space. This is something we could follow up on though – thanks for the thought.

9-26: Thanks very much. This is useful to know. Actually, we added some new material here in response to Reviewer 2. Soruco et al. (2015) found that reduced glacier area had been compensated by increased melt rates.

9-29: This is an important point, and I'm not sure that there is a clear answer in the literature about how important (or otherwise) glacial meltwater is for water supply in Bolivia (although see previous point and reference). One of the purposes of our paper is to stimulate further interest in this issue. Some studies have suggested that up to 40% of the water supply in La Paz is derived from glacial meltwater during the dry season (e.g. Vergara et al., 2007), whilst more recent estimates indicate a lower dependency of ∼15% (Soruco et al., 2015). Likewise, some studies have indicated

significant impacts of glacier decline on rural populations (e.g. Oxfam, 2009), whilst others have revealed that glacier change is not perhaps the most dominant force driving people away from rural locations (Kaenzig, 2015). Much of this information is already stated in the manuscript, and hopefully there is enough here to stimulate researchers to look more closely at this issue. We also made some changes relevant to this point in response to Reviewer 2.

10-26: This serves as a reason for keeping the information on ice-contact lake change as a figure rather than a Table (as discussed above). Whilst the headline is that there has been a reduction in the number of ice-contact lakes, the trend is rather chaotic over time. If the glaciers continue to shrink, then their perimeter too shrinks, and the potential for there to be ice-contact lakes reduces. Hence, I would guess that there will continue to be an overall reduction in ice-contact lakes, but that there could be a lot of variability along the way. Much will depend on what is revealed by glacier recession – there could be large overdeepenings under these glaciers that will become sites of lake development in the future. Studies along the lines of Frey et al. (2010) and Linsbauer et al. (2016), which attempt to derive glacier bed topographies, would be very welcome in this regard.

11-7 to 14: The ultimate end point would be that all glaciers disappear and hence that all potential for GLOFs disappears too - notwithstanding issues around using GLOF terminology in deglaciated regions as discussed above. The question of whether deglaciating landscapes are becoming more dangerous because of GLOF events and other hazards, or whether they become less hazardous over time because there is less area within which these hazards can take place is interesting. The record of GLOFs in Bolivia is also very poor, so it's hard to say what the longer term trend is. These issues vary on a site-by-site basis too. For example, if you take the example of Laguna Glaciar (Lake 23 in our inventory – see Supplementary material), this lake has got bigger and bigger through the course of the study period, and so arguably has become more dangerous as more of the overdeepening has been exposed. We also considered a similar

point made by Reviewer 1 (Wilfried Haeberli).

12-5: Yes, interesting point, and one that we encountered when writing Cook and Quincey (2015). From that study, "As yet, there is no reliable technique available for measuring lake bathymetry or volume from satellite imagery where turbidity precludes the derivation of reflectance–depth relationships (e.g. Box and Ski, 2007)". Recent work by Pope et al. (2016) developed techniques for estimating supraglacial lake depths (http://www.the-cryosphere.net/10/15/2016/tc-10-15-2016.pdf), but obviously, these are not affected to the same degree by turbidity. Lake bathymetry is a crucial measure for GLOF hazard assessments, but remains a labour-intensive, field-based measurement.

12-25: We have re-worded this, but we maintain that the lack of reporting of GLOFs in Bolivia does not necessarily mean that they haven't occurred, nor that they won't be more frequent in the future. Reviewer 1 (Wilfried Haeberli) makes a comment on our paper that also suggests that deglaciating environments could become more hazardous.

Figure 3: Interesting – good spot. We traced our Peru-Bolivia border from the National Geographic basemap layer in ArcGIS. We double-checked our mapping, and it is correct. However, we have seen maps (e.g. GoogleEarth) where the border is drawn slightly to the east.

References

Bolch, T., Buchroithner, M., Peters, J., Baessler, M., and Bajracharya, S.: Identification of glacier motion and potentially dangerous glacial lakes in the Mt. Everest region/Nepal using spaceborne imagery, Natural Hazards and Earth System Science, 8, 1329-1340, 2008.

Box, J. E. and Ski, K.: Remote sounding of Greenland supraglacial melt lakes: implications for subglacial hydraulics, J. Glaciol., 53, 257–265, 2007.

Cook, S. and Quincey, D.: Estimating the volume of Alpine glacial lakes, Earth Surface Dynamics, 3, 559, 2015.

Frey, H., Haeberli, W., Linsbauer, A., Huggel, C., and Paul, F.: A multi-level strategy for anticipating future glacier lake formation and associated hazard potentials, Natural Hazards and Earth System Sciences, 10, 339-352, 2010.

Hoffmann, D. and Weggenmann, D.: Climate Change Induced Glacier Retreat and Risk Management: Glacial Lake Outburst Floods (GLOFs) in the Apolobamba Mountain Range, Bolivia. In: Climate change and disaster risk management, Springer, 2013.

Huggel, C., Kääb, A., Haeberli, W., Teysseire, P., and Paul, F.: Remote sensing based assessment of hazards from glacier lake outbursts: a case study in the Swiss Alps, Canadian Geotechnical Journal, 39, 316-330, 2002.

Kaenzig, R.: Can glacial retreat lead to migration? A critical discussion of the impact of glacier shrinkage upon population mobility in the Bolivian Andes, Population and Environment, 36, 480-496, 2015.

Linsbauer, A., Frey, H., Haeberli, W., Machguth, H., Azam, M., and Allen, S.: Modelling glacier-bed overdeepenings and possible future lakes for the glaciers in the Himalaya—Karakoram region, Annals of Glaciology, 57, 119-130, 2016.

Oxfam: Bolivia: Climate change, poverty and adaptation. Oxfam International, Oxford, UK, 2009.

Pope, A., Scambos, T. A., Moussavi, M., Tedesco, M., Willis, M., Shean, D., and Grigsby, S.: Estimating supraglacial lake depth in West Greenland using Landsat 8 and comparison with other multispectral methods, The Cryosphere, 10, 15-27, 2016.

Soruco, A., Vincent, C., Rabatel, A., Francou, B., Thibert, E., Sicart, J. E., and Condom, T.: Contribution of glacier runoff to water resources of La Paz city, Bolivia (16 S), Annals of Glaciology, 56, 147-154, 2015.

[Figure]

Vergara, W., Deeb, A., Valencia, A., Bradley, R., Francou, B., Zarzar, A., Grünwaldt, A., and Haeussling, S.: Economic impacts of rapid glacier retreat in the Andes, Eos, Transactions American Geophysical Union, 88, 261-264, 2007.
* * *

---

## Author Comment (AC2) · 15 Sep 2016

We thank Wilfried Haeberli for his constructive comments on our manuscript, and are pleased that he views our work as welcome and interesting. We have made a number of changes to our manuscript in response to his comments, and respond to these below.

Hazard and risk aspects

A key criticism of our manuscript is that it employs a self-correlation between lake volumes and areas (Equations 2 and 3 in the original manuscript) to predict the volumes of Bolivian glacial lakes. These comments follow constructive interactive comments made by Wilfried Haeberli about Cook and Quincey (2015) (see http://www.earth-surf-dynam.net/3/559/2015/esurf-3-559-2015-discussion.html). In our original manuscript,

we sought to avoid total reliance on the widely used empirical volume-area relationship of Huggel et al. (2002) by supplementing this with a new empirical volume-area relationship derived from a more targeted and appropriate selection of lakes from the collation of data undertaken by Cook and Quincey (2015), i.e. those of a similar area to the lakes encountered in Bolivia, and restricted to moraine-dammed lakes only. Nonetheless, this did not avoid the use of a volume-area self-relationship, and the reviewer has questioned the validity of its use here. These concerns are valid.

To address this issue in the revised manuscript, we have taken the reviewer's advice by instead using the depth-area data collated by Cook and Quincey (2015) to predict mean depth for Bolivian glacial lakes. Depth can be multiplied by measured lake area to derive lake volume. We now explain this in the text where we present Equation 2, and the updated values are shown in Table 2. We have removed the V-A relationship of Huggel et al. (2002), and we have removed values from Table 2 and in the text that were derived from that relationship. The majority of these changes can now be found on P6 L5-25.

On the subject of peak discharges calculated using Equation 4 (now equation 3), the reviewer raises the point that we should emphasise that these represent worst-case values. We have now highlighted that throughout the manuscript wherever we discuss these values, and where we describe the methods used (i.e. where Equation 3 appears). E.g. P6 L24 and P8 L22-3.

The reviewer also raises the interesting point that the potential for glacial hazards to occur may not reduce as glaciers disappear. This is pertinent to the interactive comment on our manuscript by Mauri Pelto (doi:10.5194/tc-2016-140-SC1), who queried whether GLOFs represent an "emerging" threat in Bolivia. Certainly, we agree that glacial hazards, including GLOFs, could become a worsening threat to communities in Bolivia, but there are no long-term data available (at least to our knowledge) to examine any such trends. See P5 L3-5 for example (plus other edits in response to minor corrections below).

Minor corrections

1-24: changed to "contain" P1 L24

2-04: changed to "increasing atmospheric temperature" P2 L4.

2-26: we now cite these references P2 L26-7

4-15 and 5-first paragraph: We have noted on p5 L11 that the selection of this distance threshold is somewhat subjective. We have followed the precedent here of Wang et al. (2011, 2015). We made the suggested addition about permafrost thaw, and cited a new study by Rangecroft et al. (2016) that indicates almost complete permafrost disappearance in the Bolivian Andes by the 2080s (P5 L3-5).

5-9: We have added a statement about the importance of other factors in determining flood magnitude. P5 L13-14.

5-13/14: We now acknowledge that smaller lakes could still generate damaging floods. P5 L18.

5-23: We now acknowledge that floods could propagate further than 20km. P5 L28.

7-33: removed "increase" P8 L7

8-10/14: We have added a statement to emphasise that these discharges are worst case / unlikely. P6 L24 and P8 L22-3.

11-01: We were unsure what the reviewer was suggesting. Perhaps a way of decreasing word count. We have left this for now.

11-23: We now cite Linsbauer et al 2016. P12 L25.

11-28: full stop removed P12 L12

12-28/29: We felt that the conclusion section was perhaps the wrong place to discuss rock dam stability since this appears to be an issue in its own right. Instead, we elaborated on this issue on P5 L34 to P6 L2, and removed mention of rock-dammed lakes
in the conclusions. We have emphasised in the conclusions that these are worst-case scenario values of peak discharge.

Table 2: Yes. This was dealt with in an earlier point.

References

Cook, S. and Quincey, D.: Estimating the volume of Alpine glacial lakes, Earth Surface Dynamics, 3, 559, 2015.

Huggel, C., Kääb, A., Haeberli, W., Teysseire, P., and Paul, F.: Remote sensing based assessment of hazards from glacier lake outbursts: a case study in the Swiss Alps, Canadian Geotechnical Journal, 39, 316-330, 2002.

Rangecroft, S., Suggitt, A. J., Anderson, K., and Harrison, S.: Future climate warming and changes to mountain permafrost in the Bolivian Andes, Climatic Change, 137, 231-243, 2016.

Wang, W., Xiang, Y., Gao, Y., Lu, A., and Yao, T.: Rapid expansion of glacial lakes caused by climate and glacier retreat in the Central Himalayas, Hydrological Processes, 29, 859-874, 2015.

Wang, W., Yao, T., Gao, Y., Yang, X., and Kattel, D. B.: A first-order method to identify potentially dangerous glacial lakes in a region of the southeastern Tibetan Plateau, Mountain Research and Development, 31, 122-130, 2011.

---

## Author Comment (AC3) · 15 Sep 2016

Reply to reviewer 2

We thank the reviewer for their thoughtful comments on our manuscript. We are pleased that they suggest our work be published, and that they praise the approach and merit of the work. We address the specific and minor comments below.

Specific comments

We agree that care should be taken when reporting statistics about water resources and vulnerability. It is easy to regurgitate these figures and lose a sense of the science upon which they are based, and the reviewer is right to check that we are using language and statistics appropriately. The reviewer first points to a specific example

on p1-28 of the original manuscript. We should clarify that the values cited here are based on previous work, but we had failed to cite the accompanying reference (Soruco et al., 2015), which made the value look arbitrary. We have now added that reference (p1-28). We have also added further discussion of the results of Soruco et al (2015) into section 4.1 because they studied decadal changes (or lack of) in runoff to La Paz (P10 L2-5). We have added additional references (e.g. Painter, 2007) to the Introduction to back-up points that we had made (P1-29). I would argue that, overall, we have been careful and conservative in our writing – more so than in previous studies in some cases. For example, the reviewer suggests that some values are speculative without measured values, but we already state that further quantification is required (e.g. P2-2). We are also quite careful not to be alarmist – we use phrases like "La Paz... derives *some* of its electricity from hydropower generation, which depends to *some extent* on glacial meltwater generation". Compare that with the equivalent statement in Painter (2007): "La Paz is also dependent for *virtually all* of its energy supplies on hydroelectric power". One of us (Hoffmann) lives in La Paz, and knows that this latter statement is incorrect, and hence we toned-down our writing of the importance of glaciers for hydropower in Bolivia. Elsewhere, we have toned-down our language. For example, in the abstract and introduction we had said that meltwater was "vital", but now we say it is "important"(P1-1 & 25). We hope that the addition of some missing references, along with further discussion of the few papers that have quantified glacier and runoff change (e.g. Soruco et al., 2015) will satisfy the reviewer here.

The reviewer comments on our process for identifying potentially dangerous lakes. We think that it is a good idea to include a flow diagram that summarises the process of identifying such lakes. We had considered this before submitting our manuscript, but decided against it on the basis of article length and page costs. We have, however, adopted this suggestion – see Figure 2. The reviewer's suggestion that we consider ranking the lakes in some way according to their level of danger is a good one, and this is something that we intend to work on as part of the second author's PhD project. However, there are, as yet, no universally accepted or adopted methods for determining

how dangerous a lake is – should population exposure be a more important factor than lake size? Should the potential for rockfall into a lake be of greater concern than a degrading ice-cored moraine? Because this is such a complex issue, and because we simply wanted to identify lakes that are potentially dangerous (rather than measure their absolute level of danger/threat), we opted for a simpler approach – potential flood size – as the reviewer correctly identified. We have clarified in the text that we use peak discharge to order the lakes.

The reviewer comments on our error estimates. We adopted the technique outlined in Hanshaw and Bookhagen (2014), which calculates error as a function of glacier size (perimeter) and spatial resolution of the imagery. Hence, a relatively small total perimeter could yield a larger error estimate. For the most part, our glacier mapping was undertaken automatically. As reported in previous studies, automatic mapping yields errors typically between 2 to 6% (Paul et al., 2013). But again, we have been conservative here in our error estimate because we corrected some mapping manually. In reality, we expect the error to be lower than 10%. Hence, we cited Paul et al. (2013) with their error estimates for clean glaciers of ∼5%.

We are glad the reviewer likes the kmz file – we hope this will prove to be a useful resource for anyone interested in the issues raised in our manuscript.

The reviewer suggests we look into reasons why patterns of lake development over time are not straightforward. Certainly, this is something that we are interested in, and intend to follow-up on in our continuing work. But we are keen to keep this manuscript focused on the three stated objectives outlined in the Introduction. The second objective was to evaluate lake development (rather than explain it fully), which we have achieved. This is principally to allow us to move on to objective 3 to identify potentially dangerous lakes. Some interesting patterns emerge from our evaluation of lake development, but to explain these patterns could become a study in its own right. We are grateful for the suggestions and enthusiasm of the reviewer on how such a study could be approached. We will follow-up on this.
The reviewer asks us to clarify whether we consider moraine-dammed lakes to be more dangerous than bedrock-dammed lakes. As we state in section 2.3 (P5 L31), moraine-dammed lakes are generally considered to be more dangerous because there is the potential for breach incision through the moraine thickness, allowing for a greater volume of water to escape from the lake. But we also make the point that bedrock-dammed lakes can be sources of GLOFs, and cite some examples of this. So both are potentially dangerous, although there is a general consensus (as far as we are aware) that moraine-dammed lakes represent the greater threat. Hence, we categorised lakes by dam type in Table 2.

The reviewer questions our ordering or lakes in Table 2. Again, we have now clarified in the text that we have simply ranked the lakes by peak discharge (P6 L25). This is because there are no universally accepted methods of ranking lakes by threat – this is something we are working on, however.

The reviewer asks us to reconsider our use of the word "trend" in relation to ice-contact lake data, which show no strong tendency. This is a fair point, and we have adjusted our language accordingly (e.g. by using the word "change" instead). E.g. P11 L9.

Minor comments

P3 L5: We have now changed this to "initial assessment". P3 L6 & P4 L23.

P4L26: We have added a colon. P4L28

P5 L23: We weren't sure what was being commented on here, so we have kept this the same.

P5 L27: We had to change this anyway in response to another comment – it now reads "lakes confined within". P5 L34.

P8 L21: Changed as requested. P8 L29

P10 L27: Changed as requested. P11 L9

P11 L28: Changed as requested. P12 L12

P11, L32,33: Yes. Done. P12 L16-18

Map figures: We experimented with different map formats in preparing this manuscript – different ESRI backgrounds, Landsat imagery – natural and false colour. One of the issues is scale: our area of interest is very large, but the glaciers and lakes relatively small. If we were to plot up all of the locations of settlements from geobolivia for this region, it would somewhat dominate the map (essentially, a lot of dots). This is one of the reasons why we produced the supplementary kmz file so that people could look at the detail for themselves, whilst leaving the map figures fairly simple in order to illustrate glacier change and lake locations.

Table 2: Please see our response to earlier comments. We prefer to keep this very simple and order by peak discharge. Thanks for the space-saving tip. We saved a lot of space anyway in dealing with Reviewer 1's comments to remove the values derived from lake volume-area self-correlation. But we also changed the heading to "Dam type" to allow us to shorten the classifications to "moraine" and "bedrock".

---

## Author Response (AR2)

**Author's Response**

Dear Dr. Berthier,

5 I am hereby submitting a revised version of our manuscript:

Cook, S. J., Kougkoulos, I., Edwards, L. A., Dortch, J., and Hoffmann, D.: Glacier change and glacial lake outburst flood risk in the Bolivian Andes, The Cryosphere Discuss., doi:10.5194/tc-2016-140.

10

Please find below responses to the two reviewers (Reviewer 1 Wilfried Haeberli; Reviewer 2 Anonymous) and the Interactive Comment by Mauri Pelto, as well as a marked-up version of the manuscript that illustrates all of the changes that we have made. Our general sense is that all three interactive comments are very positive about the manuscript, and see it as a valuable and publishable

15 piece of research. Indeed, Mauri Pelto even wrote a blog piece for the AGU on our manuscript, indicating an enthusiasm for this work, as well as promoting *The Cryosphere*: http://blogs.agu.org/fromaglaciersperspective/2016/07/18/chaupi-orko-glaciers-bolivia-extensiverecession/

20 In general, we have accepted the comments of the reviewers and adjusted our manuscript accordingly. Indeed, we have found the review process to be very constructive – certainly, the manuscript has been improved.

If you have any further queries, need any additional changes or information from us, then please do not hesitate to contact me.

1

With best wishes,

Mall

30

Simon

Dr. Simon Cook Senior Lecturer in Physical Geography 35 Manchester Metropolitan University S.J.Cook@mmu.ac.uk +44 (0) 161 2471202

**Reply to Mauri Pelto – Interactive Comment**

We thank Mauri Pelto for his insightful comments and questions about our manuscript. This provides us with an opportunity to elaborate on some points in our study. Taking each point in turn (our responses are in red):

5

10

15

20

Cook et al: 2016 provide a detailed examination of glacier area change and glacier lakes in key glaciated regions of Bolivia. This is a valuable paper that advances both regional knowledge of glacier and glacier lake change, but also utilizes efficient and reliable techniques. The GLOF threat and the current trend in that threat may be overstated, given the current decline in ice-contact glacier lakes noted and the author's observation that few new lakes seem to be forming and the lakes will likely have reduced sizes. The impact on glacier runoff of the 43% glacier area loss is likely understated and should be better quantified, though still being a first order approximation.

We are pleased that Mauri Pelto finds this to be a valuable study. It is noteworthy that he and Reviewer 1 are in some disagreement about the potential for GLOFs in areas with reducing ice cover. We come onto this point later on. Later on, we discuss the issue of runoff – a paper by Soruco et al. (2015) is particularly instructive, as it indicates no change so far in runoff despite the dramatic glacier changes.

2-27: The lack of attention is likely due to their not being any historic experience with GLOF suggesting the risk is not high in this region. Hoffman and Weggenmann (2012) point out that the Keara example here is the first known GLOF in the area.

It's hard to say with any certainty if the GLOF risk in the region is high or not. One of the points of our paper is to highlight that there is a risk, and that this risk has not been assessed before. Hoffmann & Wegenmann (2013) described the only documented GLOF in Bolivia, as far as we are aware, but that does not necessarily mean that it is the only GLOF that has occurred here. It is worth remembering that, for the most part, the people affected by such events live in rather remote

- 25 communities. GLOF events may simply go undocumented. Last year, we travelled to Agua Blanca, in the Apolobamba Region (Northern part of our study area), where we spoke to the village leader about the work we were doing there and this led him to recount a story from his father who had witnessed a GLOF event some years ago in a neighbouring valley. There is no documentation of this event, and it is hard to verify, but the point is clear: there may be other undocumented GLOFs that have taken place in Bolivia.
- 30

4-1: The step wise approach for glacier area identification is a best practice approach. OK 4-27: If these lakes are not in contact with the glacier and the steep slope has not been recently deglaciated, than is GLOF is the correct term?

I think this is a reasonable point. I suppose the issue is whether a lake that occupies a basin that has been carved out by a glacier or dammed by glacial sediment constitutes a 'glacial lake', and hence if it were to burst can be considered a 'glacial lake

5 outburst flood'. I agree that the term 'GLOF' would normally be used for outbursts from ice-contact lakes or lakes that had formed 'recently' following deglaciation (whatever you take 'recent' to mean). I hope that most readers would appreciate what is meant when we use the term 'GLOF' in the context of our work in Bolivia.

5-1: I assume this applies to only recently deglaciated steep slopes, if not just state.

10 Not necessarily. In Bolch et al.'s (2008) paper, the whole study region is classified according to slope, with the slopes of 45 deg or steeper representing the greatest mass movement risk. They aren't necessarily recently deglaciated. We just wanted to make the extra point here that recently deglaciated slopes can be particularly susceptible to mass movements.

5-28: Reword; Lakes confined within rock basins are less likely to experience breach incision.

15 Good idea. We have reworded.

6-3: Any examples of application of this formula in the Andes?

No, not that I am aware of. In compiling data for an earlier study (Cook and Quincey, 2015), we generated a list of studies that had employed the lake volume-area relationship of Huggel et al. (2002). At the time of doing that (c. 2014-15) there were some studies that borrowed the relationship to estimate lake volume for glacial lakes in the high mountains of Asia (Nepal, Tien

20 studies that borrowed the relationship to estimate lake volume for glacial lakes in the high mountains of Asia (Nepal, Tien Shan, etc), but none for the Andes. We have subsequently removed the equation from Huggel et al. (2002) in response to comments by Reviewer 1 (Wilfried Haeberli).

7-10: This is one of the key finding that all regions had a decline in ice contact lakes from 1986 to 2014, Figure 6a does not communicate this as well as a Table would.

I suppose the extent to which a table vs. a figure best illustrates a trend in data is debateable. But I would like to emphasise a key issue here – we are not only making the point that ice-contact lakes have declined over time, but also that the pattern of change over time has been rather chaotic. I would argue that this is best illustrated with a figure because it is more immediately comprehensible.

30

25

7-17: A Table would provide a better display of lake number changes than Figure 6a or 6b, since these are not actual trends through the study period, a line chart display does not provide a clear picture of what is occurring. A table could also better quantify the total rock basin versus moraine dammed lakes.

As with the previous point, I would argue instead that the graph shows the complexities in lake change over time more effectively than a table of numbers, which would be harder to digest. I take the point that a Table would be useful in terms of information on lake type, but this is perhaps most relevant for potentially dangerous lakes – for these lakes, information on moraine-, rock-dammed lakes, etc, is provided in Table 2.

8-10: Were the number of dangerous lakes determine at any other time than 2014? If so how many were there? How many of these 25 are ice-contact?

No, we did not determine the number of dangerous lakes for other years. Our reasoning for this is that, although understanding how the GLOF risk has evolved over time would certainly be interesting, the more important issue is to determine which of

10 the lakes from the most recent dataset would be most likely to represent a GLOF risk. This seemed to us to be the more urgent question, and hence the one that should be given over to journal page space. This is something we could follow up on though – thanks for the thought.

9-26: Can emphasize this point with some quantification. A 43% decline in glacier area suggests that total glacier runoff would

- 15 have already declined substantially as runoff is product of glacier area and ablation rate. This will also affect timing as noted by below studies. "Despite a 15% increase in ablation rate, the 45% decline in glacier area led to a 38% decrease in glacier runoff in the Skykomish River basin" (Pelto, 2011). For specifics on the tropical Andes and seasonal impacts, Vuille et al. (2008) is useful. Some areas with losses of 20% of glaciated area have documented declines in glacier runoff and timing changes, that will be occurring here as well Stahl and Moore (2006), Dery et al. (2009) and Pelto (2011).
- 20 Thanks very much. This is useful to know. Actually, we added some new material here in response to Reviewer 2. Soruco et al. (2015) found that reduced glacier area had been compensated by increased melt rates.

9-29: Is glacier water supply change felt keenly? This is a region without significant hydropower or fisheries; hence the impact would be on municipal and agriculture impacts. This is an impact with much wider potential importance than GLOF's and

25 deserves further attention. Provide some information on annual precipitation in the region beyond the glaciers and some understanding of the relative water supply from glaciers to this area. A sentence or two defending the statement would prove valuable.

This is an important point, and I'm not sure that there is a clear answer in the literature about how important (or otherwise) glacial meltwater is for water supply in Bolivia (although see previous point and reference). One of the purposes of our paper

30 is to stimulate further interest in this issue. Some studies have suggested that up to 40% of the water supply in La Paz is derived from glacial meltwater during the dry season (e.g. Vergara et al., 2007), whilst more recent estimates indicate a lower dependency of ~15% (Soruco et al., 2015). Likewise, some studies have indicated significant impacts of glacier decline on rural populations (e.g. Oxfam, 2009), whilst others have revealed that glacier change is not perhaps the most dominant force driving people away from rural locations (Kaenzig, 2015). Much of this information is already stated in the manuscript, and

5

hopefully there is enough here to stimulate researchers to look more closely at this issue. We also made some changes relevant to this point in response to Reviewer 2.

10-26: The decline in ice-contact lakes suggests the number of pro-glacial lakes will decline as glacier retreat continues in the near future. Is this true?

5

This serves as a reason for keeping the information on ice-contact lake change as a figure (Fig 6) rather than a Table (as discussed above). Whilst the headline is that there has been a reduction in the number of ice-contact lakes, the trend is rather chaotic over time. If the glaciers continue to shrink, then their perimeter too shrinks, and the potential for there to be ice-contact lakes reduces. Hence, I would guess that there will continue to be an overall reduction in ice-contact lakes, but that

- 10 there could be a lot of variability along the way. Much will depend on what is revealed by glacier recession there could be large overdeepenings under these glaciers that will become sites of lake development in the future. Studies along the lines of Frey et al. (2010) and Linsbauer et al. (2016), which attempt to derive glacier bed topographies, would be very welcome in this regard.
- 15 11-7 to 11-14: This discussion illustrates that GLOF risk should be declining as the authors note fewer new lake basins are being exposed or likely to be exposed and their size is reduced. This should be emphasized in abstract and conclusion. The ultimate end point would be that all glaciers disappear and hence that all potential for GLOFs disappears too. The question of whether deglaciating landscapes are becoming more dangerous because of GLOF events and other hazards, or whether they become less hazardous over time because there is less area within which these hazards can take place is interesting. The record
- 20 of GLOFs in Bolivia is also very poor, so it's hard to say what the longer term trend is. These issues vary on a site-by-site basis too. For example, if you take the example of Laguna Glaciar (Lake 23 in our inventory see Supplementary material), this lake has got bigger and bigger through the course of the study period, and so arguably has become more dangerous as more of the overdeepening has been exposed. We also considered a similar point made by Reviewer 1 (Wilfried Haeberli).
- 25 12-5: Is there any chance in your view that remote sensing could be used to assess lake depth in combination with some ground truth? If so elaborate, if not leave alone.

Yes, interesting point, and one that we encountered when writing Cook and Quincey (2015). From that study, "As yet, there is no reliable technique available for measuring lake bathymetry or volume from satellite imagery where turbidity precludes the derivation of reflectance–depth relationships (e.g. Box and Ski, 2007)". Recent work by Pope et al. (2016) developed

30 techniques for estimating supraglacial lake depths (http://www.the-cryosphere.net/10/15/2016/tc-10-15-2016.pdf), but obviously, these are not affected to the same degree by turbidity. Lake bathymetry is a crucial measure for GLOF hazard assessments, but remains a labour-intensive, field-based measurement.

12-25: Given the statements on 2-30 indicating a lack of previous damaging GLOF's and the decline in ice-contact lakes, I do not see how emerging can be used. If anything the data here suggests GLOF's risk will decline. It is certainly appropriate to emphasize the prior lack of quantification of the risk.

We have re-worded this, but we maintain that the lack of reporting of GLOFs in Bolivia does not necessarily mean that they
haven't occurred, nor that they won't be more frequent in the future. Reviewer 1 (Wilfried Haeberli) makes a comment on our paper that also suggests that deglaciating environments could become more hazardous.

**Figure 3: Is the Bolivia Peru Border in the Chaupi Orko Massif too far east?**

Interesting – good spot. We traced our Peru-Bolivia border from the National Geographic basemap layer in ArcGIS. We
double-checked our mapping, and it is correct. However, we have seen maps (e.g. GoogleEarth) where the border is drawn slightly to the east.

5

**Reply to Reviewer 1 - Wilfried Haeberli**

We thank Wilfried Haeberli for his constructive comments on our manuscript, and are pleased that he views our work as welcome and interesting. We have made a number of changes to our manuscript in response to his comments, and respond to

**these below (in red).**

General

5

In agreement with the editorial comments by Etienne Berthier and the comments by Mauri Pelto I consider this contribution to be a welcome and interesting study about a less well-documented regional development. As the two mentioned colleagues discussed the questions of remote sensing, glacier mapping and hydrology, my comments can focus on questions of hazard

10 and risk assessments. My recommendations aim at encouraging the authors to more critically reflect their techniques and formulations concerning people-related hazards and risks. Scientific hazard and risk studies are policy-relevant and require aspects of transparency and honesty to be especially critically reflected.

Hazard and risk aspects

- 15 Equations (2) and (3) are used for area-based estimates of lake volumes, which are then applied as input for calculating peakdischarge values, which in turn are taken as indicators of hazard potentials. There are three fundamental problems related to the application of such unfortunately quite popular equations: Volume-area relations are unnecessary area-area self-relations, the preciseness of the numerical values used in the regression equations is disproportionate with respect to the accuracy and reliability of the results obtained and the statistics deal with mean instead of extreme values which is an uncommon and 20 problematic procedure in hazard consideration.
- Lake volumes are determined by multiplying measured lake areas with measured and averaged/integrated lake depths. Correlating lake volume with lake area, therefore, means to correlate a mathematical product with one of the factors from which it had been calculated. Correspondingly, predicting lake volumes from lake areas means to essentially predict lake areas as used for volume calculation from themselves. Why should we do this? Because "many do it" (including myself in earlier
- 25 papers)? Because the statistics and scatter plots look better? Or because the knowledge on how lake volumes are determined is lost? I strongly recommend to strictly avoid unnecessary volume-area self-relations but to use the straightforward relation between the originally determined lake areas and lake depths. This straightforward approach produces exactly the same results but is transparent and honest in that it not only shows what is measured and what is calculated but especially also illustrates the large scatter in the relation between the two measured variables and shows the resulting enormous uncertainty in the
- 30 estimated values as well as in all further values (here volumes, peak discharges) derived from them. In fact, the morphometric analysis of glacier-bed overdeepenings where lakes may form clearly shows that large features can be shallow and small features can be deep (Haeberli et al. 2016). This should be clearly said in order to avoid unrealistic expectations: Orders of magnitude for lake volumes can at best be estimated empirically. Correspondingly, the excess preciseness of the numerical values used in equations (2) and (3) derived from statistical regression and used in such order-of-magnitude estimates is

disproportionate and provides a misleading impression of the accuracy and reliability which can be reached in reality. In addition to these two problems, equations (2) and (3) also have the problem that they represent medium-value statistics while hazard assessment must be made with extreme-value statistics for worst-case considerations.

- 5 A key criticism of our manuscript is that it employs a self-correlation between lake volumes and areas (Equations 2 and 3 in the original manuscript) to predict the volumes of Bolivian glacial lakes. These comments follow constructive interactive comments made by Wilfried Haeberli about Cook and Quincey (2015) (see <a href="http://www.earth-surf-dynam.net/3/559/2015/esurf-3-559-2015-discussion.html">http://www.earth-surf-dynam.net/3/559/2015/esurf-3-559-2015-discussion.html</a>). In our original manuscript, we sought to avoid total reliance on the widely used empirical volume-area relationship of Huggel et al. (2002) by supplementing this with a new empirical volume-10 area relationship derived from a more targeted and appropriate selection of lakes from the collation of data undertaken by Cook and Ouineau (2015), i.e. these of a similar area to the lakes anequatered in Bolivian and restricted to maxima dammed lakes.
- and Quincey (2015), i.e. those of a similar area to the lakes encountered in Bolivia, and restricted to moraine-dammed lakes only. Nonetheless, this has not avoided the use of a volume-area self-relationship, and the reviewer has questioned the validity of its use here. These concerns are valid.
- 15 To address this issue in the revised manuscript, we have taken the reviewer's advice by instead using the depth-area data collated by Cook and Quincey (2015) to predict mean depth for Bolivian glacial lakes. Depth can be multiplied by measured lake area to derive lake volume. We now explain this in the text where we present Equation 2, and the updated values are shown in Table 2. We have removed the V-A relationship of Huggel et al. (2002), and we have removed values from Table 2 and in the text that were derived from that relationship. The majority of these changes can now be found on P6 L5-25.

20

25

Equation (4) is an example and outcome of such reflections: It avoids oversophisticated and over-precise mean-value and selfrelation statistics but enables in a perfectly transparent way the realistic estimation of empirical extremes in a simple way even without any computer: the fact that the right-hand side of this equation is written as 2V/1000 instead of V/500 helps to make its application as easy as possible, even in the field or for non-scientists. This simplicity and transparency also makes it clear to scientists as well as to stakeholders or even the public where the limits are of our knowledge, understanding and ability to

- predict numbers of unmeasured lake volumes for practical applications in the real world. Extreme peak-discharge values as estimated, for instance, by equation (4) refer to worst-case events. In the case of glacial and periglacial lakes, such extreme peak values can result from sudden-break mechanisms of dams consisting of broken ice from ice avalanches or glacier surges, from massive erosion and debris-flow formation in connection with moraine breaching, or
- 30 from squeezing-out of more or less entire lakes by large ice/rock avalanches. This is again essential to be made clear. Worstcase scenarios with such extreme peak discharge values relate to low-frequency/highmagnitude events. Dealing with corresponding hazards from events with extremely low probability but also with extreme damage potential are a special challenge for policy making with regard to risk acceptance and risk management. Most outburst processes such as, for instance, progressive enlargement of sub-glacial channels produce far smaller peak discharges. Careful wording of worst-case scenarios is necessary

in order to avoid adverse psychological and economic effects which can exceed the potential damage occurring in reality or perhaps even not occurring at all.

On the subject of peak discharges calculated using Equation 4 (now equation 3), the reviewer raises the point that we should emphasise that these represent worst-case values. We have now highlighted that throughout the manuscript wherever we discuss these values, and where we describe the methods used (i.e. where Equation 3 appears). E.g. P6 L24 and P8 L22-3.

5

The continued glacier retreat indeed tends to reduce glacier sizes and, hence, areas and volumes of new lakes forming as a consequence of ice vanishing. It also reduces direct ice-contacts of lake water with calving fronts. This does, however, not necessarily reduce the hazard potential. The more the glaciers retreat the closer new lakes form to steep walls of icy peaks with

- degrading permafrost and long-term reduction of slope stability. The corresponding scientific literature is easily accessible 10 today. In Bolivia, permafrost can be expected at altitudes above about 5000m a.s.l where the 0 C isotherm is found (cf. Carey et al. 2012). This may be a lesser influence on the hazard situation in the investigated region but must be correctly mentioned and treated. De-buttressed slopes and slopes with degrading permafrost are the two situations with the most rapid and important change in long-term rockfall disposition.
- 15 The reviewer also raises the interesting point that the potential for glacial hazards to occur may not reduce as glaciers disappear. This is pertinent to the interactive comment on our manuscript by Mauri Pelto (doi:10.5194/tc-2016-140-SC1), who queried whether GLOFs represent an "emerging" threat in Bolivia. Certainly, we agree that glacial hazards, including GLOFs, could become a worsening threat to communities in Bolivia, but there are no long-term data available (at least to our knowledge) to examine any such trends. See P5 L3-5 for example (plus other edits in response to minor corrections below).
- 20

Minor corrections

1-24: "contain" not contains. Changed to "contain" P1 L24

2-04: The term "climate warming" is popular but not scientifically correct and should better be replaced by something like "climate change", "global warming" or "atmospheric temperature increase": Climate is defined as an average of meteorological 25 parameters (not only temperature) over extended time periods (usually decades). As such it cannot "warm" (the permafrost can warm). Changed to "increasing atmospheric temperature" P2 L4.

2-26: Carey et al. (2012) and Haeberli et al. (2016; as mentioned in the text) could be added here concerning the recent example of Laguna 513 in the Cordillera Blanca). We now cite these references P2 L26-7

4-15 and 5-first paragraph: The 500m limit is highly subjective and not really reliable: Ice avalanches over firn/ice surfaces

can have trajectory lengths 3 times the drop height and rock or rock/ice avalanches can by far exceed such limits. If the authors 30 prefer to stick to their number they should comment on it accordingly. The phenomenon of permafrost should be mentioned here. We have noted on p5 L11 that the selection of this distance threshold is somewhat subjective. We have followed the precedent here of Wang et al. (2011, 2015). We made the suggested addition about permafrost thaw, and cited a new study by

Rangecroft et al. (2016) that indicates almost complete permafrost disappearance in the Bolivian Andes by the 2080s (P5 L3-5).

5-9: The involved processes of outburst triggering and flood propagation in the torrent below the lakes are of primary importance concerning potential damages and corresponding hazard potentials, rather than lake area or lake volume. This

5 should be mentioned. We have added a statement about the importance of other factors in determining flood magnitude. P5 L13-14.

5-13/14: This assumption is again delicate: squeezing out of a small lake by an ice or rock avalanche may cause far-reaching floods with high damage potential. We now acknowledge that smaller lakes could still generate damaging floods. P5 L18.

5-23: Trajectory slopes have so far been used rather than simple distances. Process chains can affect infrastructure over much

10 longer distances than 20 km. Again: A problem of low frequency/high magnitude events. Most events will remain far within distances of 20 km but in the worst case 20 km may by far not be enough. We now acknowledge that floods could propagate further than 20km. P5 L28.

7-33: Eliminate "increase" after 72%. Removed "increase" P8 L7

8-10/14: make clear that these values refer to highly unprobable worst-case scenarios. We have added a statement to emphasise
 that these discharges are worst case / unlikely. P6 L24 and P8 L22-3.

11-01: ": : : : decreased in such cases : : :"? We were unsure what the reviewer was suggesting. Perhaps a way of decreasing word count. We have left this for now.

11-23: a more recent reference would be Linsbauer et al. (2016). We now cite Linsbauer et al 2016. P12 L25.

11-28: eliminate one full stop after ": : : infrastructure." Full stop removed P12 L12

20 12-28/29: mention that these values are worst-case scenarios and add that the "higher stability" of lakes in rock basins may be questioned in case of possible impact waves produced by rock/ice avalanches from de-buttressed slopes or slopes with degrading permafrost (cf. especially Deline et al. 2014). We felt that the conclusion section was perhaps the wrong place to discuss rock dam stability since this appears to be an issue in its own right. Instead, we elaborated on this issue on P5 L34 to P6 L2, and removed mention of rock-dammed lakes in the conclusions. We have emphasised in the conclusions that these are

25 worst-case scenario values of peak discharge.

Table 2: Lake volumes are not extreme values but peak discharges are extreme values from worst-case scenarios. At least point to this discrepancy and discuss it in the text. Yes. This was dealt with in an earlier point.

**Reply to Reviewer 2**

We thank the reviewer for their thoughtful comments on our manuscript. We are pleased that they suggest our work be published, and that they praise the approach and merit of the work. We address the specific and minor comments below (in

5 red)

**General comments:**

This paper fills a gap identified in the literature for an updated inventory of glacier mass changes and formation of potentially dangerous lakes in Bolivia. There are relatively few glaciers and a total glacier lake area of less than 10 km2 in Bolivia. Still,

- 10 apparently there has not been a comprehensive and up-to-date accounting of glacier and lake changes for all ranges assessed here. This paper nicely provides a time trajectory of change over the most recent decades, when it has been widely recognized that change is occurring at unprecedented rates. The observations are therefore inherently valuable and should be published. The paper is well-written and clear, and the methods of acquisition sound. However, the scientific merit of the data analyses is less obvious in terms of advancing process understanding from the basic observations (i.e. this is clearly a reporting of
- 15 important observations; what are the implications in terms of our understanding of dangerous lake formation?).

**Specific comments**

Care should be taken in how related issues of water resources and vulnerability are included. It should be clarified that without measured values, the discussion of water supply and urban migration remains speculative. For example, what is the basis for

- 20 estimating 15% of potable water to El Alto and La Paz come from glaciers (P1-28)? And is this water originating from a net loss of glacier mass, or is it simply from glacier fed streams? Without specifying, these general statements amount to speculative hyperbole. The authors should provide numbers, and references. Similarly, much of La Paz power comes from hydroelectrity? How much power generation is really vulnerable to being unreliable with low flows?
- We agree that care should be taken when reporting statistics about water resources and vulnerability. It is easy to regurgitate 25 these figures and lose a sense of the science upon which they are based, and the reviewer is right to check that we are using language and statistics appropriately. The reviewer first points to a specific example on p1-28 of the original manuscript. We should clarify that the values cited here are based on previous work, but we had failed to cite the accompanying reference (Soruco et al., 2015), which made the value look arbitrary. We have now added that reference (p1-28). We have also added further discussion of the results of Soruco et al (2015) into section 4.1 because they studied decadal changes (or lack of) in
- 30 runoff to La Paz (P10 L2-5). We have added additional references (e.g. Painter, 2007) to the Introduction to back-up points that we had made (P1-29). I would argue that, overall, we have been careful and conservative in our writing more so than in previous studies in some cases. For example, the reviewer suggests that some values are speculative without measured values, but we already state that further quantification is required (e.g. P2-2). We are also quite careful not to be alarmist we use phrases like "La Paz... derives \*some\* of its electricity from hydropower generation, which depends to \*some extent\* on

glacial meltwater generation". Compare that with the equivalent statement in Painter (2007): "La Paz is also dependent for \*virtually all\* of its energy supplies on hydroelectric power". One of us (Hoffmann) lives in La Paz, and knows that this latter statement is untrue, and hence we toned-down our writing of the importance of glaciers for hydropower in Bolivia. Elsewhere, we have toned-down our language. For example, in the abstract and introduction we had said that meltwater was "vital", but now we say it is "important" (P1-1 & 25). We hope that the addition of some missing references, along with further discussion of the few papers that have quantified glacier and runoff change (e.g. Soruco et al., 2015) will satisfy the reviewer here.

There is a sequence of steps taken to both automatically analyze the scenes, and also use human expertise to evaluate "dangerous lakes" vis-a-vis factors of risk and damage potential. Why not present this algorithm as a flow diagram? And why

10 not provide relative scale/magnitude of danger? It is interesting to consider the area of lakes that meet the threshold of 'dangerous'. Why not present a graduated ranking of significance from least to most dangerous? The inventory of lakes seems to be numbered from smallest to largest volume/Qmax. But even that is not explicit in the text. The reviewer comments on our process for identifying potentially dangerous lakes. We think that it is a good idea to include

a flow diagram that summarises the process of identifying such lakes. We had considered this before submitting our manuscript, but decided against it on the basis of article length and page costs. We have, however, adopted this suggestion –

- see Figure 2. The reviewer's suggestion that we consider ranking the lakes in some way according to their level of danger is a good one, and this is something that we intend to work on as part of the second author's PhD project. However, there are, as yet, no universally accepted or adopted methods for determining how dangerous a lake is should population exposure be a more important factor than lake size? Should the potential for rockfall into a lake be of greater concern than a degrading ice-
- 20 cored moraine? Because this is such a complex issue, and because we simply wanted to identify lakes that are potentially dangerous (rather than measure their absolute level of danger/threat), we opted for a simpler approach potential flood size as the reviewer correctly identified. We have clarified in the text that we use peak discharge to order the lakes.

Error is equated to uncertainty, determined to be 10%. Is this rather high? This is based on an assumption of Gaussian distribution. I'm just unclear about how this relates to the reporting of areas.

The reviewer comments on our error estimates. We adopted the technique outlined in Hanshaw and Bookhagen (2014), which calculates error as a function of glacier size (perimeter) and spatial resolution of the imagery. Hence, a relatively small total perimeter could yield a larger error estimate. For the most part, our glacier mapping was undertaken automatically. As reported in previous studies, automatic mapping yields errors typically between 2 to 6% (Paul et al., 2013). But again, we have been

30 conservative here in our error estimate because we corrected some mapping manually. In reality, we expect the error to be lower than 10%. Hence, we cited Paul et al. (2013) with their error estimates for clean glaciers of ~5%.

I do like the inclusion of a kmz file for viewing the lakes. Simple and effective.

5

25

We are glad the reviewer likes the kmz file – we hope this will prove to be a useful resource for anyone interested in the issues raised in our manuscript.

In assessing the lake formation related to glacier change, it seems that more analyses of basic variables could be easily explored

5 to substantiate some of the patterns that the authors observe, and help make more robust suggestions about processes. I think it most compelling to explore pattern of lake changes that might be anticipated given the valley morphology. For example, how do patterns of lake changes relate to topographic indices (e.g. hypsometry, lake elevation, distance from headwall)? What about relating lake changes to glacier forms? These seems straight forward derivatives of this impressive database that has been generated. Then, it would seem the authors might articulate more clear hypotheses to explain the lack of trend in ice-10 contact lake formation over time. How do these patterns compare to other regions?

The reviewer suggests we look into reasons why patterns of lake development over time are not straightforward. Certainly, this is something that we are interested in, and intend to follow-up on in our continuing work. But we are keen to keep this manuscript focused on the three stated objectives outlined in the Introduction. The second objective was to evaluate lake development, which we have achieved. This is principally to allow us to move on to objective 3 to identify potentially

15 dangerous lakes. Some interesting patterns emerge from our evaluation of lake development, but to explain these patterns could become a study in its own right. We are grateful for the suggestions and enthusiasm of the reviewer on how such a study could be approached. We will follow-up on this.

It is not clear: were moraine-dammed lakes considered more dangerous or not? If there is no difference, why bother categorizing them as such? Explain.

The reviewer asks us to clarify whether we consider moraine-dammed lakes to be more dangerous than bedrock-dammed lakes. As we state in section 2.3 (P5 L31), moraine-dammed lakes are generally considered to be more dangerous because there is the potential for breach incision through the moraine thickness, allowing for a greater volume of water to escape from the lake. But we also make the point that bedrock-dammed lakes can be sources of GLOFs, and cite some examples of this.

25 So both are potentially dangerous, although there is a general consensus (as far as we are aware) that moraine-dammed lakes represent the greater threat. Hence, we categorised lakes by dam type in Table 2.

I wonder why the lakes are numbered as they are (by size of Q max, starting from smallest?). Why not by region? Or keeping #1 as the largest (i.e. reflecting risk magnitude)?

30 The reviewer questions our ordering or lakes in Table 2. Again, we have now clarified in the text that we have simply ranked the lakes by peak discharge (P6 L25). This is because there are no universally accepted methods of ranking lakes by threat – this is something we are working on, however. Throughout this discussion, it is probably misleading to talk about "trend" with respect to the change in ice-contact lakes since there is not a significant tendency to the data.

The reviewer asks us to reconsider our use of the word "trend" in relation to ice-contact lake data, which show no strong tendency. This is a fair point, and we have adjusted our language accordingly (e.g. by using the word "change" instead). E.g. P11 L9.

**Minor Comments**

5

10

15

P3 L5: what is meant by "first-pass assessment"? Is it spelled with dash or not (compare P4L23)? We have now changed this to "initial assessment". P3 L6 & P4 L23.

P4L26: (and elsewhere): use colon before numbered list. We have added a colon. P4L28

P5 L23: "Direct hydrological connection" to downstream infrastructure and communities. We weren't sure what was being commented on here, so we have kept this the same.

P5 L27: Lakes "sat" should be set. We had to change this anyway in response to another comment – it now reads "lakes confined within". P5 L34.

P8 L21: use 'glacierized' for currently glacier covered, as glaciated can refer to previously ice covered. Changed as requested. P8 L29

P10 L27: "very variable" is awkward phrase; better to use "highly variable". Changed as requested. P11 L9 P11 L28: double periods. Changed as requested. P12 L12

20 P11, L32,33: remove the parentheses as this is a substantive point. Yes. Done. P12 L16-18

Map figures: the ESRI hillshade backdrop is okay, but they lack of any downstream features (population centers, roads) that are used in evaluating the 'dangerous' lake conditions. We experimented with different map formats in preparing this manuscript – different ESRI backgrounds, Landsat imagery – natural and false colour. One of the issues is scale: our area of interest is very large, but the glaciers and lakes relatively small. If we were to plot up all of the locations of settlements from

25 geobolivia for this region, it would somewhat dominate the map (essentially, a lot of dots). This is one of the reasons we produced the supplementary kmz file so that people could look at the detail for themselves, whilst leaving the map figures fairly simple in order to illustrate glacier change and lake locations.

Table 2: separate lake number and list by range. Could also abbreviate lake type to save space. Please see our response to earlier comments. We prefer to keep this very simple and order by peak discharge. Thanks for the space-saving tip. We saved

30 a lot of space anyway in dealing with Reviewer 1's comments to remove the values derived from lake volume-area self-correlation. But we also changed the heading to "Dam type" to allow us to shorten the classifications to "moraine" and "bedrock".

[revised manuscript text omitted]

- 10 Where V is lake volume in m3, and A is lake surface area in m2. Cook and Quincey (2015) noted highlighted some issues in this approach, that the relationship of Huggel et al. (2002) performed well in estimating lake volumes in most cases, but that there were some situations where lakes could be especially shallow or deep, giving unusually small or large volumes for a given area. Hence, they advocated that geomorphological context be considered when deciding on which empirical approach to adopt. All of the lakes identified in our inventory are either moraine-dammed, or sit within rock basins. Specifically, volume-
- 15 area relationships suffer from auto-correlation between volume and area, which gives an unrealistic impression of the strength of correlation between the two. Lake volume is typically calculated by multiplication between measured lake area and averaged or integrated lake depth. Consequently, it is more appropriate to use depth and area data, which can be measured independently, to estimate lake volume. Cook and Quincey (2015) also suggested that geomorphological context be considered when applying empirical techniques to estimate volume because different lake types (i.e. ice-dammed, moraine-dammed, thermokarst, etc.)
- 20 can have different morphological characteristics. All of the lakes identified in our inventory are either moraine-dammed, or sit within rock basins. Given these suggestions, we used the dataset from Cook and Quincey (2015) to derive an empirical relationship between lake depth and area that is specific to moraine-dammed lakes of a similar area range to those found in this studyThe relationship of Huggel et al. (2002) was shown by Cook and Quincey (2015) to perform well for estimating the volume of moraine-dammed lakes because the data used to generate Equation 1 were derived largely from moraine-dammed lakes. For those cases, we adopt Equation 1, but also use the larger dataset from Cook and Quincey (2015) to derive an
- empirical relationship specific to moraine dammed lakes of a similar area range to those found in this study. This takes the form:

 $\frac{\Psi D}{(32)} = 0.097 A^{\pm 0.4375} \qquad -$

30 Where D is mean lake depth in m, and A is lake surface area in m2. Equation 2 can be used to predict mean lake depth, which can be multiplied by measured lake area to calculate lake volume. We are not aware of any empirical formula for predict the volume estimation where of lakes are situated within rock basins. We are not aware of any empirical formula formula for the volume estimation where of lakes are situated within rock basins. We are not aware of any empirical formula formula for the volume estimation where of lakes are situated within rock basins. We are not aware of any empirical formula formula for the volume estimation where of lakes are situated within rock basins.

volume estimation where lakes are situated within rock basins. In the absence of any such formula, we use Equation Equation s 1 and 22 to provide a first order estimation of their volumes as well as moraine-dammed lakes.

Lake volume can be used to estimate peak discharge ( $Q_{max}$ ). Huggel et al. (2002) collated several empirical models for estimating GLOF peak discharge from lake volume, but ultimately adapted the relationship of Haeberli (1983) to give:

$$Q_{max} = \frac{2V}{t} \tag{43}$$

Where t, time, is equal to 1000 seconds. We used equation 3 to estimate peak discharge for the lakes identified as being potentially dangerous. It should be noted that peak discharges calculated using Equation 3 represent worst-case estimates. Lakes are ranked from lowest to highest peak discharges in Table 2.

**10 3 Results**

30

5

**3.1 Glacier change 1986-2014**

Our results reveal that total glacier areal cover across the Bolivian Cordillera Oriental in 1986 was  $529.3 \pm 52.9$  km2, and that by 2014 this area had reduced to  $301.2 \pm 30.1$  km2 (Figure 2Figure 3a). This represents a total areal reduction of 43.1 % over the 28-year study period. If the Peruvian Cordillera Apolobamba glaciers are included in the dataset, then the total glacier

- 15 cover in 1986 was  $626.5 \pm 62.7 \text{ km}^2$  and in 2014 it was  $351.7 \pm 35.2 \text{ km}^2$ , representing a 43.9 % reduction. Figure 3/2 Figure 3/a illustrates the reduction in overall glacier cover across this period. Rates of ice loss appear to vary across the study period, with an initially rapid shrinkage between 1986 and 1992 (14.5 km2 a-1), relatively modest losses between 1992 and 1999 (5.1 km2 a-1), strong ice shrinkage between 1999 and 2010 (8.1 km2 a-1), and modest losses between 2010 and 2014 (4.0 km2 a-1) (except for the Tres Cruces region).
- For consistency with earlier studies, we present results in Figure 2Figure 3 of glacier areal change for separate glaciated mountain ranges, and Figures 3-4 to 5-6 illustrate glacier change as a series of maps for each region. All mountain ranges show decreases in overall glacier area across the study period with a total loss of 43.1 % glacier cover in the Bolivian Cordillera Apolobamba (172.3 ± 17.2 km2 to 96.0 ± 9.6 km2), 41.9% across the Cordillera Real (315.2 ± 31.5 km2 to 183.1 ± 18.3 km2), and 47.3 % in the Tres Cruces region (41.8 ± 4.2 km2 to 22.0 ± 2.2 km2). 
[revised manuscript text omitted]